# Real-time transcriptomic profiling in distinct experimental conditions

**Tamer Butto[1]\*[†][‡], Stefan Pastore[1,2][†], Max Müller[1], Kaushik Viswanathan Iyer[1], Marko Jörg[1], Julia Brechtel[1], Stefan Mündnich[1], Anna Wierczeiko[2], Kristina Friedland[1], Mark Helm[1], Marie-Luise Winz[1], Susanne Gerber[2]\*[‡]**

[1]Institute of Pharmaceutical and Biomedical Sciences, Johannes Gutenberg-University Mainz, Mainz, Germany; [2]Institute of Human Genetics, University Medical Center of the Johannes Gutenberg University Mainz, Mainz, Germany

---

## eLife Assessment

This **useful** study presents a real-time transcriptomics analysis, with the aim of providing rapid access to sequenced data to reduce the costs associated with Oxford Nanopore long-read technology. The revised manuscript demonstrates the utilities with four sets of experiments with **convincing** evidence.

---

**\*For correspondence:**
buttamer@uni-mainz.de (TB);
sugerber@uni-mainz.de (SG)

[†]These authors contributed equally to this work

[‡]Joint senior authors

**Competing interest:** The authors declare that no competing interests exist.

**Abstract** Nanopore technology offers real-time sequencing opportunities, providing rapid access to sequenced data and allowing researchers to manage the sequencing process efficiently, resulting in cost-effective strategies. Here, we present focused case studies demonstrating the versatility of real-time transcriptomics analysis in rapid quality control for long-read RNA-seq. We illustrate its utility through four experimental setups: (1) transcriptome profiling of distinct human cellular populations, (2) identification of experimentally enriched transcripts, (3) transcriptional analysis of cells under heat shock conditions, and (4) identification of experimentally manipulated genes (knockout and overexpression) in several yeast strains. We show how to perform multiple layers of quality control as soon as sequencing has started, addressing both the quality of the experimental and sequencing traits. Real-time quality control measures assess sample/condition variability and determine the number of identified genes per sample/condition. Furthermore, real-time differential gene/transcript expression analysis can be conducted at various time points post-sequencing initiation (PSI), revealing dynamic changes in gene/transcript expression between two conditions. Using real-time analysis, which occurs in parallel to the sequencing run, we identified differentially expressed genes/transcripts as early as 1 hr PSI. These changes were consistently observed throughout the entire sequencing process. We discuss the new possibilities offered by real-time data analysis, which have the potential to serve as a valuable tool for rapid and cost-effective quality checks in specific experimental settings and can be potentially integrated into clinical applications in the future.

---

## Introduction

The field of transcriptomics aims to explore, monitor, and quantify the complete set of transcripts, including coding (e.g. mRNA), non-coding, and small RNAs, within a given cell at a given condition (*Wang et al., 2009*). The investigation of the transcriptome is crucial for understanding the functional elements of the genome and their role within a cell or tissue, as well as their role during development or disease manifestation (*Casamassimi et al., 2017*). Over the past decade, transcriptomics has witnessed significant technological advancements, especially with the rise of Next Generation Sequencing (NGS) and the extensive use of RNA sequencing (RNA-seq; *Mutz et al., 2013*; *Satam*

*et al., 2023*). Techniques such as RNA-seq became the primary methodology to investigate the transcriptome using high-accuracy, short-read data (*Mutz et al., 2013*; *Satam et al., 2023*; *Butto et al., 2023*). Additionally, several well-established bioinformatic pipelines for RNA-seq have demonstrated reliability in analyzing transcriptome data. These pipelines typically involve quality control, read alignment to a reference genome, quantification of gene expression levels, and downstream analysis of differential gene expression. Notable tools such as minimap2 (*Li, 2018*), HISAT2 (*Kim et al., 2019*), or STAR (*Dobin et al., 2013*) are commonly employed for read alignment, while featureCounts (*Liao et al., 2014*) or HTSeq (*Anders et al., 2015*) are utilized for quantifying expression. The widely used DESeq2 (*Love et al., 2014*) and edgeR (*Robinson et al., 2010*) packages offer robust statistical methods for identifying differentially expressed genes. The reliability of such tools is evidenced by their widespread adoption in the scientific community, which allows the extraction of meaningful insights from RNA-seq data and contributes to our understanding of gene expression dynamics in various biological contexts (*Conesa et al., 2016*; *Ji and Sadreyev, 2018*; *Corchete et al., 2020*).

While RNA-seq coupled with NGS has revolutionized transcriptome analysis, there are still improvements to be made, mainly depending on the requirements of the experimental design. The costs associated with NGS RNA-seq experiments can be a considerable factor, particularly when dealing with a large number of samples or the experimental approach requires high sequencing depth (*Conesa et al., 2016*; *Ji and Sadreyev, 2018*). For instance, higher read depth often yields more comprehensive information (e.g. splicing/isoform detection analyses; *Zhang et al., 2017*; *Hardwick et al., 2019*). However, this comes at the expense of higher costs. Secondly, the library preparation process for NGS RNA-seq poses inherent challenges since it involves fragmentation of the reverse transcribed cDNA and introducing potential PCR bias during library amplification (*Ozsolak and Milos, 2011*). These steps may introduce a limitation in accurately representing the investigated transcriptome, as certain sequences might be preferentially amplified over others, ultimately resulting in the loss of valuable information. Lastly, repetitive sequences pose a significant obstacle in their analysis, especially when employing short-read sequencing technologies (*Ozsolak and Milos, 2011*). For instance, the precise alignment of short reads to repeat regions/elements remains problematic due to the intrinsic nature of such reads (*Ozsolak and Milos, 2011*). Thus, it is essential to consider alternative sequencing strategies to address these obstacles.

One noteworthy alternative is long-read sequencing, such as Nanopore sequencing (Nanopore-seq). This technology, developed by Oxford Nanopore Technologies (ONT), has emerged as an innovative method for sequencing native long-read nucleic acids, including genomic DNA, cDNA, and RNA (*Lu et al., 2016*; *Wang et al., 2021*; *Zheng et al., 2023*). The library preparation procedure involves straightforward steps, integrating a specific adapter at the end of the nucleic acid. This facilitates the efficient 'reading' of intact nucleic acids, even ultra-long fragments (*Kono and Arakawa, 2019*; *Amarasinghe et al., 2020*). Integrating long-read sequencing with transcriptomics allows for the capture of entire transcripts, providing distinct advantages in detecting various RNA isoforms, repetitive sequences, and long mRNA transcripts (*Amarasinghe et al., 2020*; *Wang et al., 2021*). In addition, one key advantage of Nanopore-seq lies in its capability of real-time sequencing. This feature provides the opportunity to gain rapid access to the sequenced data, enabling researchers to either manage the sequencing process or stop it once the desired results are achieved (*Wang et al., 2021*). The latter allows for washing and reusing consumables, thus significantly lowering the sequencing costs. Moreover, adaptive sampling offers opportunities to enrich or deplete specific genes or transcripts during runtime (*Wang et al., 2024*). Direct RNA sequencing facilitates the detection of various RNA modifications on the basis of characteristic raw signal divergences. Modification detection in combination with real-time sequencing is becoming increasingly important for both basic and clinical research (*Alagna et al., 2025*; *Brändl et al., 2025*; *Hewel et al., 2024*; *Pastore et al., 2025*; *Spangenberg et al., 2025*). A few studies have reported real-time analysis tools coupled with Nanopore-seq, primarily focusing on genomic or metagenomic DNA applications. For instance, real-time analysis platforms like EPI2ME by ONT (https://labs.epi2me.io/) and minoTour (*Munro et al., 2022*) provide continuous access to real-time metrics and analysis, streamlining the sequencing process. Algorithmic tools such as BOSS-RUNS (*Weilguny et al., 2023*), RawHash (*Firtina et al., 2023*), and BoardION (*Bruno et al., 2021*) introduce dynamic decision strategies, hash-based similarity searches for efficient real-time analysis, and interactive web applications for ONT sequencing runs. Additional real-time detection tools, such as Metagenomic (*Sanderson et al., 2018*) and NanoRTax (*Rodríguez-Pérez et al., 2022*),

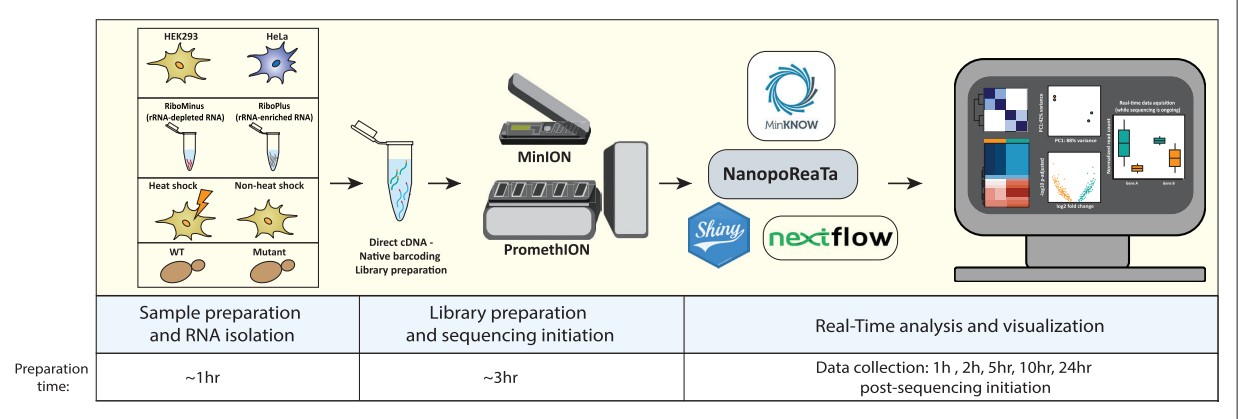

**Figure 1.** Streamlined pipeline for the rapid experimental setup and utilization of NanopoReaTA. Sample preparation involved Trizol RNA isolation (~1 hr) following library preparation, which included the synthesis of dscDNA library for the desired RNA sample (~2 hr). Samples were barcoded, and adapter ligated (~1 hr). Sequencing was performed using a PromethION R10 flow cell or MinION R9 flow cell (for HEK293 and HeLa), and real-time data analysis and visualization occurred alongside ongoing sequencing. For this study, figures were exported at 1 hr, 2 hr, 5 hr, 10 hr, and 24 hr psi, providing insights into the dynamic transcriptional changes of long-read RNA-seq between distinct conditions.

provide immediate analytical pathways, concentrating on assessing metagenomic composition and viral detection tools. This diverse array of tools collectively addresses various aspects of Nanopore sequencing, spanning real-time analysis, algorithmic enhancements, metagenomic exploration, and current signal mapping. However, the combination of real-time analysis alongside comprehensive transcriptomic analysis has not been extensively explored.

Recently, we presented NanopoReaTA, the first real-time analysis toolbox for comparative transcriptional analyses of Nanopore-seq data (*Wierczeiko et al., 2023*). NanopoReaTA provides an interactive graphic user interface (GUI) that allows users to perform transcriptional analyses of cDNA and/or direct RNA libraries. The new possibilities offered by real-time analysis are precious for fast and cost-effective quality control. In addition, they have the potential to significantly impact clinical applications where speed and efficiency are crucial, for example in diagnostics. Here, we present streamlined case studies, demonstrating the utility of real-time analysis using NanopoReaTA in various rapid quality control layers.

## Results
### Experimental design

We designed four experimental setups that include: (1) transcriptome profiling of distinct human cellular populations, (2) identification of experimentally enriched transcripts, (3) transcriptional profiling of cells exposed to heat shock, and (4) identification of an experimentally manipulated gene (KO and overexpression) in yeast strains (*Figure 1*). The latter demonstrates that real-time analysis using NanopoReaTA can also be applied to non-mammalian samples, provided the genome annotation files are available. We have designed a streamlined pipeline (experimental and bioinformatic) to monitor the detection speed of the transcriptional changes occurring between distinct conditions. We simultaneously performed a pairwise comparison between two distinct conditions (Appendix 1 - 'Step-by-step use of NanopoReaTA'). According to the capabilities of the computational device (256 GB RAM) and the size of the reference genome (Human ~40 GB RAM, Yeast ~8 GB RAM) in use, multiple instances of NanopoReaTA were used in parallel. We set up five data collection time points from the sequencing initiation, including 1 hr, 2 hr, 5 hr, 10 hr, and 24 hr post-sequencing initiation (PSI). While sequencing, we exported several analyzed datasets, including general sample overviews such as read length distribution (per sample and condition), gene expression variability (per sample and condition), changes in gene composition (per sample and condition), and processing time.

Additionally, we performed analyses of real-time differential gene/transcript expression (DGE/DTE) and differential transcript usage (DTU) between the two conditions, providing valuable quality control for the experimental setup. Analyses of DGE and DTE were performed by DESeq2 (*Love*

*et al., 2014*), which is integrated into NanopoReaTA's pipeline. For DTU, we integrated analysis tools such as DEXSeq (*Anders et al., 2012*) and DRIMSeq (*Nowicka and Robinson, 2016*). This feature offers insights into specific transcript isoforms differentially expressed between distinct conditions. All output tables and figures produced by NanopoReaTA were systematically gathered and arranged to track the real-time detection of transcriptional changes during sequencing.

## Efficient segregation of distinct cellular populations using NanopoReaTA's rapid transcriptome profiling

To demonstrate the rapidity and precision of real-time analysis in detecting transcriptional changes, we chose two distinct cell populations with unique transcriptomes and monitored alterations while sequencing was in progress. HEK293 (Human Embryonic Kidney) and HeLa (cancer) cells were selected due to their simplicity, widespread use, and ease of manipulation. To simulate distinct experimental designs, we structured three different setups. In the first, we employed 10 biological replicates per cell type, providing enhanced reliability and precision of the statistical analysis, reducing the impact of variability and allowing for more accurate identification of significant differences. The second setup simulated a scenario with 2 replicates per condition, focusing on testing the effect of limited replication while generating high-throughput data using PromethION. The final setup compared the performance of 2 replicates per condition using a MinION, simulating an early-stage evaluation of the experimental approach. Following barcoding, the samples were loaded into a PromethION or MinION flow cells and sequenced for 24 hr. We tracked the basic sequencing metrics using ONT's MinKnow software and activated NanopoReaTA as soon as the sequencing initiated.

One-hour post-sequencing initiation (PSI), we gathered basic sequencing metrics from the MinKNOW software, including total reads generated per sample, along with mapped reads, gene counts, and transcript counts (*Figure 2—figure supplement 1*, *Figure 2—source data 1*). At this stage, we gathered basic quality control information, including the number of detected genes, gene variability, individual and combined read length distribution, and the usage timings of the tools applied by NanopoReaTA (*Figure 2—figure supplements 2–4*). We monitored the amount of cDNA generated and loaded for sequencing, observing relatively consistent throughput between HEK293 and HeLa (*Figure 2—figure supplements 2–4*). Detailed descriptions of the experimental setup are provided in the Appendix.

Next, we performed real-time DGE and DTE analyses to monitor the transcriptional changes between HEK293 and HeLa, 1 hr PSI. As an initial quality control, we inspected the Sample-to-sample similarity plot and principal component analysis (PCA). For the 10-replicate PromethION experimental setup, we observed a separation between the two conditions, where PC1 represents 28% of the variance while PC2 represents 5% of the variance, already 1 hr PSI (*Figure 2B-C*, *Figure 2—figure supplement 5A-B*). Similar observations were noted in the 2-replicate PromethION experimental setup with distinct separation between the two conditions, where PC1 represents 68% of the variance while PC2 represents 17% of the variance (*Figure 2—figure supplement 6A–B*). Real-time measurements like these offer valuable quality control insights into experimental replicates' and conditions' quality, influencing the decision to continue sequencing based on their clustering.

In the next phase, we inspected the differentially expressed genes (DEGs) presented in the output volcano plots and the top 20 DEGs (based on fold change and adjusted p-value; *Figure 2D*, *Figure 2—source data 1*). Notably, in the 10 replicate per condition setup, we identified 23 annotated genes enriched in HEK293 and 35 genes enriched in HeLa, 1 hr PSI (*Figure 2D*, *Figure 2—figure supplement 5C*). Early comparison with 2 replicate per condition setup revealed genes such as *ACTB*, *FTL*, and *S100A6* enriched in HeLa, whereas mitochondrial rRNA genes such as *MT-RNR2* and *MT-RNR1* were enriched in HEK293 (*Figure 2—figure supplements 5C-D and 6C-D*). In addition, NanopoReaTA offers an interactive utility enabling users to input specific genes and visualize both raw and normalized read counts. Using the 'Gene-wise' utility, we introduced several of the DEGs as input genes and visualized their raw and normalized gene counts (*Figure 2—figure supplements 5E and 6E*). This application proves valuable for monitoring specific genes of interest such as cell-specific marker genes.

Following that, we conducted a DTE analysis using Salmon based read counts (*Patro et al., 2017*) in contrast to the DGE analysis using featureCounts (*Liao et al., 2014*) based read counts. It is acknowledged that employing diverse analysis tools may yield varying numbers and specific

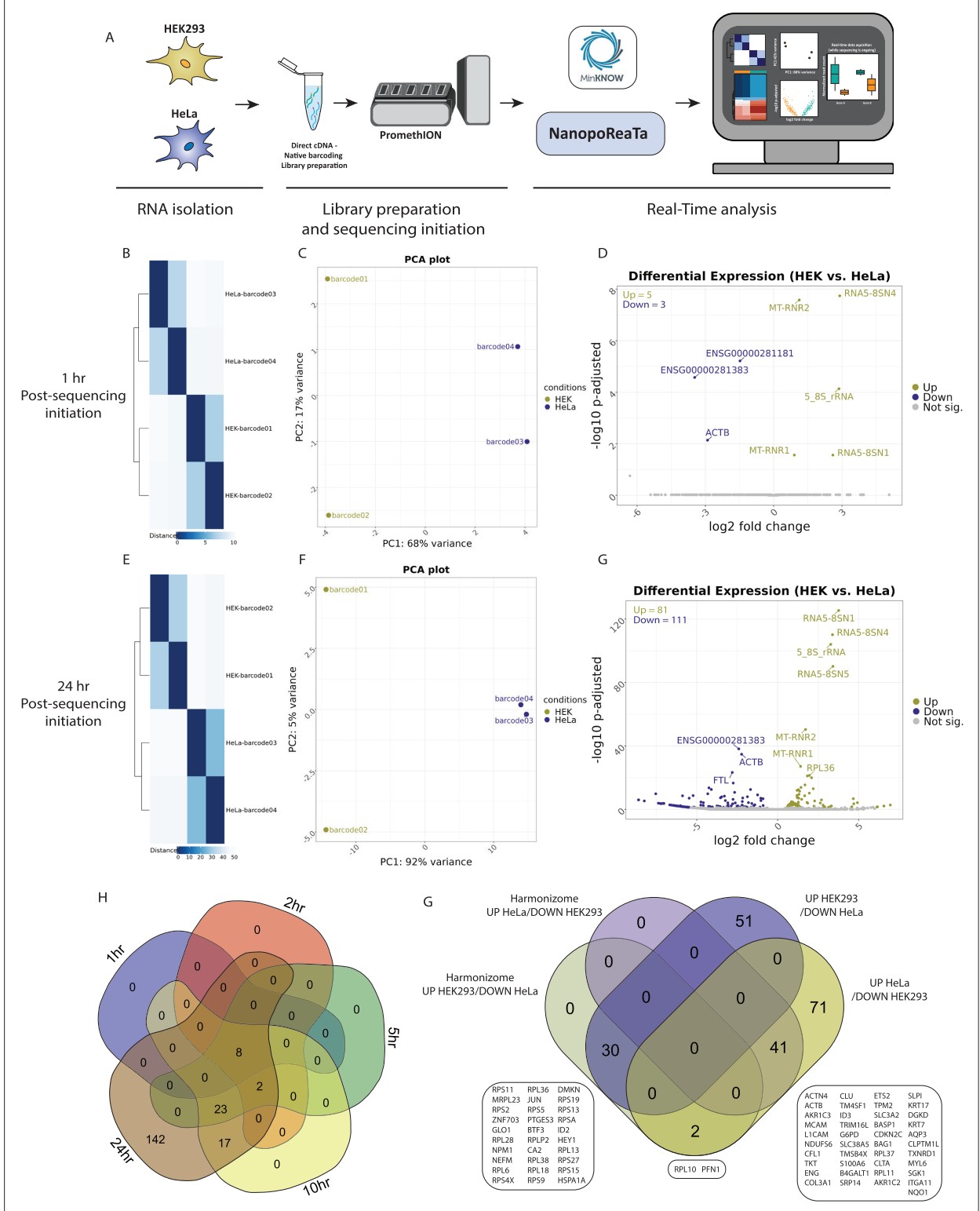

**Figure 2.** Real-time transcriptomic analysis between HEK293 and HeLa using NanopoReaTA. (**A**) Experimental strategy. RNA was isolated from HEK293 and HeLa cells, and the dscDNA library was prepared, which included sample barcoding and adapter ligation. Samples were loaded and sequenced using a PromethION R10 flow cell. NanopoReaTA was activated shortly after sequencing initiation and data was collected 1 hr, 2 hr, 5 hr, 10 hr, and 24 hr post-sequencing initiation. (**B–D**) Differential gene expression 1 hr post-sequencing initiation between HEK293 and HeLa samples. Selected data plots showing sample-to-sample distance plot (**B**), Principal component analysis (PCA) (**C**), and volcano plot (**D**) 1 hr post-sequencing initiation. (**E–G**) Differential gene expression 24 hr post-sequencing initiation between HEK293 and HeLa samples. Selected data plots showing sample-to-sample

*Figure 2 continued on next page*

*Figure 2 continued*

distance plot (**E**), PCA (**F**), and volcano plot (**G**) 24 hr post-sequencing initiation. (**H**) Five-way Venn diagram showing the differentially expressed gene overlaps between the distinct collected time points. (**I**) Validation of identified differentially expressed genes through Harmonizome database (*Rouillard et al., 2016*). For this analysis, we utilized 'HPA Cell Line Gene Expression Profiles' (*Uhlén et al., 2015*). Among the DEGs (identified at the 24 hr time point), 19 genes were found to be enriched in HEK293 and depleted in HeLa, while 46 genes showed enrichment in HeLa and depletion in HEK293. One gene (PFN1) was enriched in HeLa and enriched in HEK293 Harmonizome database.

The online version of this article includes the following source data and figure supplement(s) for figure 2:

**Source data 1.** General sequencing overview of HEK293 and HeLa (2 and 10 replicates per condition).

**Figure supplement 1.** Experimental strategy of HEK293 versus HeLa and sequencing overview.

**Figure supplement 2.** PromethION (10 rep) - General overview in HEK293 vs HeLa.

**Figure supplement 3.** PromethION (2 rep) - general overview in HEK293 vs HeLa.

**Figure supplement 4.** MinION - general overview HEK293 vs HeLa.

**Figure supplement 5.** PromethION (10 rep) - gene expression analysis in HEK293 vs HeLa.

**Figure supplement 6.** PromethION - gene expression analysis in HEK293 vs HeLa.

**Figure supplement 7.** MinION – gene and transcript expression in HEK293 vs HeLa.

**Figure supplement 8.** Transcript expression in HEK293 vs HeLa (10 rep).

**Figure supplement 9.** Transcript expression in HEK293 vs HeLa.

**Figure supplement 10.** Comparative gene expression analysis between HEK293 and HeLa cells across PromethION and MinION sequencing platforms.

differentially expressed genes or transcripts. Thus, we intended to include several established tools, ensuring that significant results are consistently identified across different methods. DTE analyses offered by NanopoReaTA generate similar visual representations to DGE, including PCA, Sample-to-sample distance plots, volcano plots, and heatmaps (*Figure 2—figure supplements 8A-D and 9A-D*). Therefore, when examining the differentially expressed transcripts (DETs) in the 10-replicate setup at 1 hr PSI, we detected 24 transcripts enriched in HEK293 and 36 transcripts enriched in HeLa (*Figure 2—figure supplement 8C–D*). In the 2 replicate per condition setup, we detected two transcripts enriched in HEK293 and six transcripts enriched in HeLa (*Figure 2—figure supplement 9C–D*). Subsequently, as data was collected at 2 hr, 5 hr, 10 hr, and 24 hr PSI, our objective was to compare the entire dataset and thus provide a dynamic real-time view of the RNA sequencing run. As expected, a noticeable increase in the number of identified genes (*Figure 2—figure supplements 2–4*), as well as DEGs and DETs (*Figure 2—figure supplements 5C-D, 6C-D, 7A-H, 8C-D, 9C-D*), was observed with the advancement of sequencing. This analysis demonstrates the ability to capture real-time changes in gene expression between distinct conditions, thus providing a valuable quality control measure.

At 24 hr PSI, we examined the sample-to-sample distance plots and PCA plots and noted enhanced separation between the conditions, particularly evident in PC2, which accounted for 88% of the variance (*Figure 2E and F*, *Figure 2—figure supplement 5A*). A similar trend was detected in 2 replicate per condition setup (*Figure 2—figure supplement 6A*). This clustering trend persisted consistently throughout the entire sequencing process, evident in both DGE and DTE analyses (*Figure 2—figure supplements 5A and 6A*). Ultimately, we identified in a 10-replicate setup, 2036 genes enriched in HEK293 and 1399 genes enriched in HeLa (*Figure 2G*, *Figure 2—source data 1*).

To assess the consistency of the results provided by NanopoReaTA, we cross-referenced the total differentially expressed genes identified at each time point, providing insights into the dynamic changes detected throughout sequencing. In the 10-replicate setup, we identified 56 annotated genes that were consistently detected across all time points, with an increased number of DEGs detected from 1 hr to 24 hr PSI (*Figure 2H*). These observations highlight the dynamic detection of DEGs during the ongoing sequencing process, emphasizing that distinctions in the most abundant transcripts likely emerge early after sequencing initiation.

Lastly, we overlapped DEGs across various experimental setups to assess the reproducibility of transcriptomic profiling between HEK293 and HeLa cells. Here, we identified 16 DEGs that were consistently detected in HEK293 and 51 in HeLa across setups (*Figure 2—figure supplement 10A–B*). Notably, MinION sequencing showed fewer unique DEGs compared to PromethION due to its lower throughput but demonstrated consistent enrichment patterns within each condition. To validate that the DEGs correspond to each condition, we utilized the Harmonizome database (*Rouillard et al.,*

*2016*) which contains a collection of datasets that compares the differential gene expression across different cell lines ('HPA Cell Line Gene Expression Profiles', *Uhlén et al., 2015*). We selected DEGs that overlapped across at least two experimental setups (47 upregulated in HEK293 and 114 upregulated in HeLa) and cross-referenced them with Harmonizome expression profiles. This analysis confirmed consistent enrichment patterns across our experimental setups (*Figure 2—figure supplement 10C–D*), identifying 19 DEGs shared in HEK293 and 46 in HeLa (*Figure 2—figure supplement 10D*). Notably, genes such as *RPS19* and *RPL18* were specifically enriched in HEK293 compared to HeLa, while *ACTB*, *CLU*, and *ID3* were enriched in HeLa compared to HEK293 (*Figure 2—figure supplement 10C–D*). These findings highlight the robustness of NanopoReaTA in detecting transcriptional differences across platforms and conditions, demonstrating its utility for real-time differential gene expression analysis.

## Real-time analysis provides rapid identification of experimentally enriched transcripts

Next, we aimed to assess the rapid detection capabilities of NanopoReaTA for experimental-enriched transcripts. To achieve this, we performed ribosomal depletion using Ribominus rRNA depletion (Thermo Fisher Scientific, K2561) on the previously tested HEK293 samples (*Figure 3A*, *Figure 3—figure supplement 1A*). Different fractions of enriched transcripts, including ribosomal-depleted transcripts (Ribominus/RiboM) and the depleted rRNA (Riboplus/RiboP), were collected, along with total RNA (TotalR) from HEK293 as a control (*Figure 3—figure supplement 1B*). We tested three comparisons including: totalR vs RiboM, totalR vs RiboP, and RiboM vs RiboP. Two replicates per condition were barcoded, and the samples were sequenced on a PromethION flow cell for 24 hr, with data collected at the same intervals as mentioned earlier (*Figure 1*).

As for the previous experiment, 1 hr PSI, we gathered basic sequencing metrics from condition comparisons, including total reads generated per sample, along with mapped reads, gene counts, and transcript counts provided by NanopoReaTA (*Figure 3—figure supplement 1C–F*, *Figure 3—source data 1*). Interestingly, despite generating more total reads and identifying more mapped reads for TotalR and RiboP, we observed a higher number of identified gene and transcript counts in RiboM compared to the other two conditions (*Figure 3—figure supplement 1D*). These findings are intriguing, especially considering the lower amount of cDNA generated and loaded for RiboM compared to TotalR and RiboP. We discuss potential reasons for these observations in the Appendix (Appendix 2). Additionally, like in the previous section, we gathered basic quality control information, including the number of detected genes, gene variability, individual and combined read length distribution, and the usage timings of the tools applied by NanopoReaTA for quality control (*Figure 3—figure supplement 2*, *5 and 8*).

Next, we inspected the PCA and dissimilarity plots (*Figure 3*, *Figure 3—figure supplements 3A-B and 4A-B*; *Figure 3—figure supplements 6A-B and 7A-B*; *Figure 3—figure supplements 9A-B and 10A-B*). We noticed clear distinctions between all three condition comparisons, evident as early as 1 hr PSI. PC1 represented 67% of the variance between RiboM vs TotalR (*Figure 3B*), 57% of the variance between RiboP vs TotalR (*Figure 3I*) and 72% of the variance between RiboP vs RiboM (*Figure 3P*). On the other hand, PC2 represented 20%, 29% and 19% of the variance in the respective comparison. In terms of detected DEGs 1 hr PSI, we identified two annotated genes enriched and three depleted in RiboM compared to TotalR (*Figure 3C*, *Figure 3—source data 1*), five enriched in RiboP compared to TotalR (*Figure 3J*) and seven enriched and one depleted in RiboP compared to RiboM (*Figure 3Q*, *Figure 3—figure supplement 3C-D*, *6C-D and 9C-D*). Notably, the enriched annotated genes in RiboP and totalR compared to RiboM align with rRNA-related transcripts, which are predominantly enriched in RiboP (*Figure 3F, M and T*, *Figure 3—figure supplement 3E*; *Figure 3—figure supplement 6E and 9E*). When examining the DETs, we identified transcripts associated with the respective rRNA enrichment groups (*Figure 3—figure supplement 4A-D*, *7A-D and 10A-D*).

Similarly, we collected all the metrics corresponding to the sample/condition variability, annotated genes, and differentially expressed genes/transcripts for 2 hr, 5 hr, 10 hr, and 24 hr PSI. At the 24 hr PSI time point, we observed a further separation of samples in the PCA according to their conditions (*Figure 3D, K and R*). Additionally, to test further the transcripts enrichment procedure, we monitored the number of identified genes in the last collection time points between the different conditions. Interestingly, by the conclusion of the 24 hr period PSI, we identified an average of 5627 genes

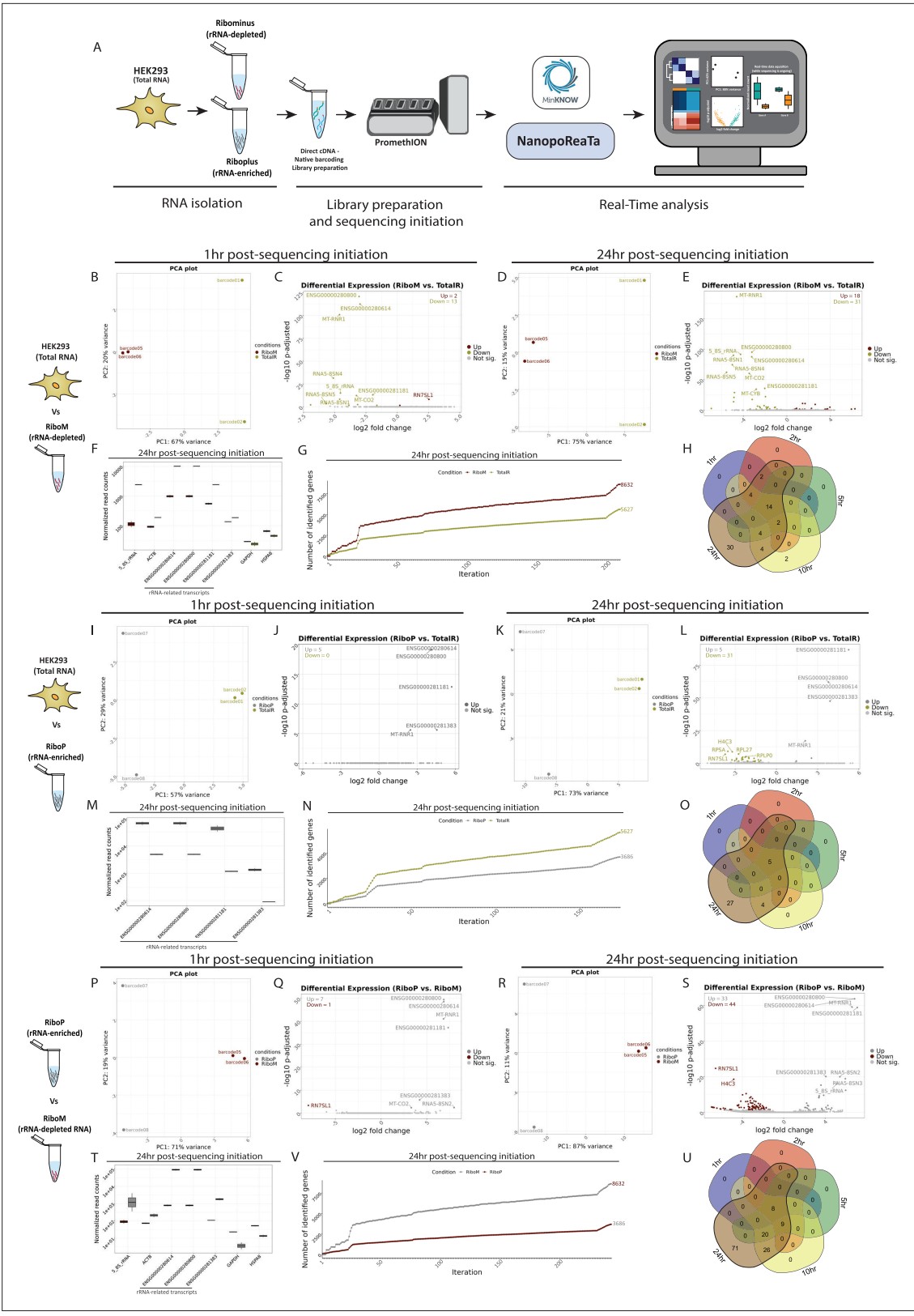

**Figure 3.** Real-time transcriptomic analysis between rRNA-depleted and rRNA-enriched transcripts using NanopoReaTA. (**A**) Experimental strategy. RNA was isolated from HEK293 and selective purification of distinct RNA populations was performed using the Ribominus Eukaryote kit. The dscDNA library was prepared, which included sample barcoding and adapter ligation. Samples were loaded and sequenced using a PromethION R10 flow cell. NanopoReaTA was activated shortly after sequencing initiation and data was collected 1 hr, 2 hr, 5 hr, 10 hr, and 24 hr post-sequencing initiation.

*Figure 3 continued on next page*

*Figure 3 continued*

(**B–H**) Differential gene expression between RiboM and TotalR from HEK293. Selected data plots showing PCA and volcano plots 1 hr (**B–C**) and 24 hr (**D–E**) post sequencing initiation. (**F**) Normalized gene counts for selected genes 24 hr. Normalized gene counts are visualized for selected genes per condition using boxplots. The median-of-ratios normalization method from DESeq2 was used for normalization. (**G**) Number of identified genes with >0 aligned reads after each iteration per condition. (**H**) Five-way Venn diagram showing the differentially expressed gene overlaps between the distinct collected time points. (**I–O**) Differential gene expression between RiboP and TotalR. Similar analyses to B-H were conducted for comparisons at 1 hr (**I–J**) and 24 hr (**K–L**) as well as normalized gene counts (**M**), gene expression variability (**N**) and Venn diagram (**O**). (**P–U**) Differential gene expression compared between RiboM and RiboP. Similar analyses to B-H were conducted for comparisons at 1 hr (**P–Q**) and 24 hr (**R–S**) as well as normalized gene counts (**T**), gene expression variability (**V**), and Venn diagram (**U**).

The online version of this article includes the following source data and figure supplement(s) for figure 3:

**Source data 1.** General sequencing overview of HEK293 Total RNA, RiboPlus and Ribominus samples (2 replicates per condition).

**Figure supplement 1.** Selective purification of ribosomal-depleted (RiboMinus) and ribosomal-enriched (RiboPlus) transcripts and sequencing overview.

**Figure supplement 2.** General overview in RiboMinus vs total RNA.

**Figure supplement 3.** Gene expression analysis in RiboMinus vs total RNA.

**Figure supplement 4.** Transcript expression in RiboMinus vs total RNA.

**Figure supplement 5.** General overview in RiboPlus vs total RNA.

**Figure supplement 6.** Gene expression analysis in RiboPlus vs total RNA.

**Figure supplement 7.** Transcript expression in RiboPlus vs total RNA.

**Figure supplement 8.** General overview in RiboPlus vs RiboMinus.

**Figure supplement 9.** Gene expression analysis in RiboPlus vs RiboMinus.

**Figure supplement 10.** Transcript expression in RiboPlus vs RiboMinus.

in TotalR, 8632 genes in RiboM, and 3686 genes in RiboP (*Figure 3G, N and V*, *Figure 3—figure supplements 2C-D and 5C-D* and *8C-D*). While these results align with our expectations, given that RiboM is strongly depleted from ribosomal RNA and RiboP consists primarily of rRNA transcripts, the depletion of rRNA in the RiboM samples may have facilitated a more efficient enrichment of non-rRNA transcripts during the double-strand cDNA synthesis procedure. Consequently, this resulted in a higher number of detected genes compared to TotalR as well, while RiboP exhibited the fewest detected genes, as anticipated.

Lastly, we overlapped the total differentially expressed genes identified at each time point to test the reproducibility of the changes detected throughout sequencing. Here, we identified 14 annotated genes in RiboM vs TotalR (*Figure 3H*), 5 annotated genes in RiboP vs TotalR (*Figure 3O*), and 8 annotated genes in RiboP vs RiboM that were consistently detected across all time points (*Figure 3U*). Interestingly, we noted an enrichment of mitochondrial rRNA in RiboP samples, which had previously been reported as depleted within the Ribominus Eukaryote Kit, thereby reinforcing the robustness of our experimental design (*Qu et al., 2013*). Overall, these findings highlight further the swift detection capabilities among transcript-enriched samples, serving as a valuable quality control measure for the rapid identification of ribosomal-depleted or polyA enrichment strategies.

## Real-time monitoring of transcriptional changes under heat shock stress

To create an experimental setup that mimics biological discovery conditions, we conducted a heat shock experiment using HEK293 cells and analyzed the resulting transcriptional changes with NanoPoReaTA. We employed six biological replicates for each condition: heat shock (HS) and non-heat shock control (NHS), using a ribodepletion protocol to enhance the capture of mRNA transcripts linked to heat-shock induction (*Figure 4A*, *Figure 4—figure supplement 1A*).

First, we gathered basic sequencing metrics from the MinKNOW software, including the total reads generated per sample, along with mapped reads, gene counts, and transcript counts (*Figure 4—figure supplement 1B–E*, *Figure 4—source data 1*). Basic quality control metrics were also collected, such as the number of detected genes, gene variability, individual and combined read length distributions, and usage timings of the tools applied by NanoPoReaTA (*Figure 4—figure supplement 2*, *Figure 4—source data 1*).

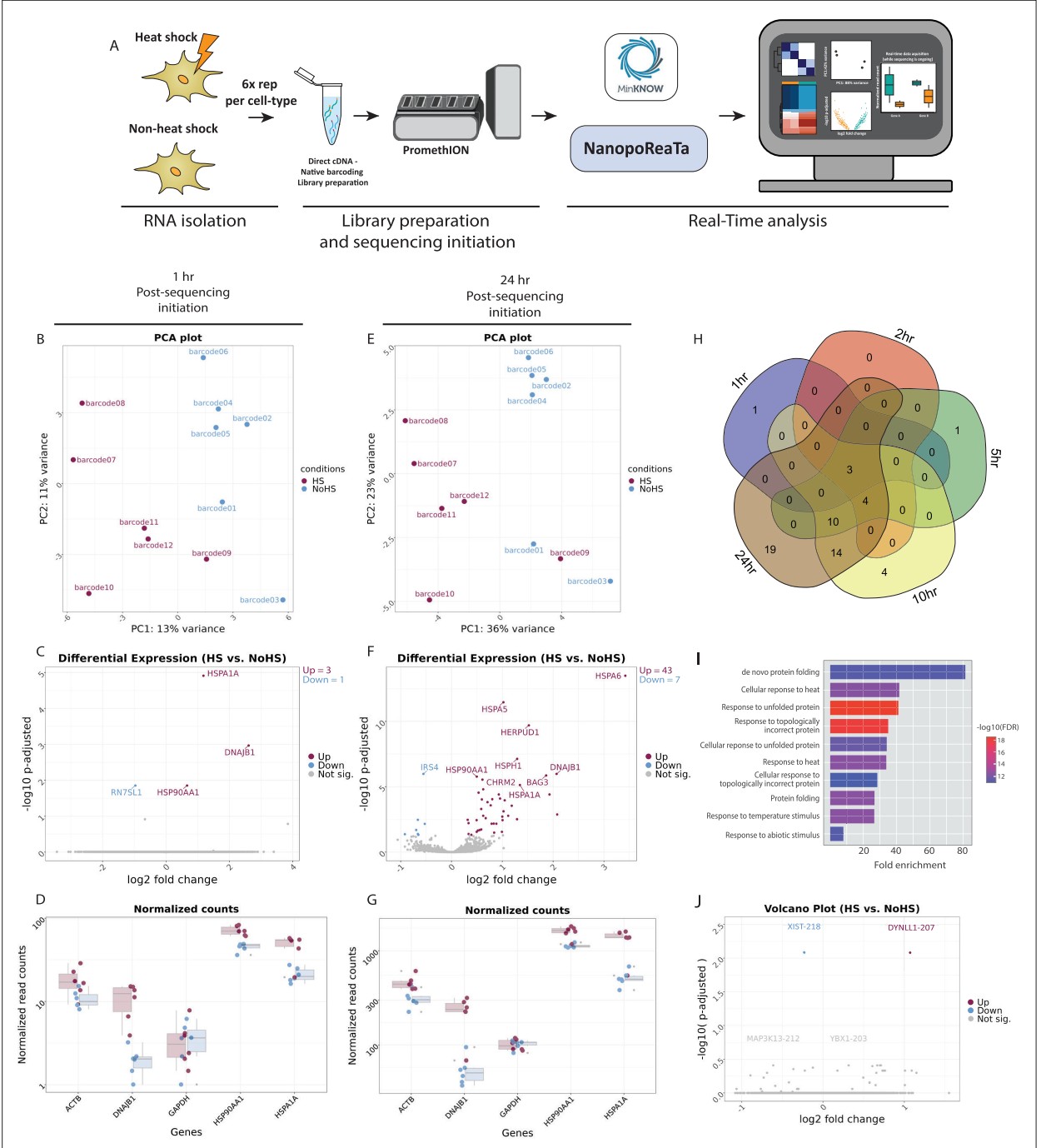

**Figure 4.** Real-time transcriptomic analysis between heat-shock and non-heat shock treated cells using NanopoReaTA. (**A**) Experimental strategy. RNA was isolated from HEK293 treated with heat shock on non-heat shock treatment (n=6), and selective purification of distinct RNA populations was performed using the Ribominus Eukaryote kit. The dscDNA library was prepared, which included sample barcoding and adapter ligation. Samples were loaded and sequenced using a PromethION R10 flow cell. NanopoReaTA was activated shortly after sequencing initiation and data was collected 1 hr, 2 hr, 5 hr, 10 hr, and 24 hr post-sequencing initiation. (**B–D**) Differential gene expression 1 hr post-sequencing initiation between HEK293 and HeLa samples. Selected data plots showing PCA (**B**), volcano plot (**C**), and normalized counts for selected genes (**D**) 1 hr post-sequencing initiation. (**E–G**) Differential gene expression 24 hr post-sequencing initiation between HEK293 and HeLa samples. Selected data plots showing PCA (**E**), volcano plot (**F**), and normalized counts for selected genes (**G**) 24 hr post-sequencing initiation. (**H**) Five-way Venn diagram showing the differentially expressed gene overlaps between the distinct collected time points. (**I**) Gene Ontology (GO) enrichment analysis of upregulated gene in HS compared to NHs conditions. (**J**) Volcano plot depicting differential transcript usage between HS and NHS conditions.

The online version of this article includes the following source data and figure supplement(s) for figure 4:

*Figure 4 continued on next page*

*Figure 4 continued*

**Source data 1.** General sequencing overview of heat-shock versus non-heat shock samples (6 replicates per condition).

**Figure supplement 1.** Experimental strategy of heat-shock experiments and sequencing overview.

**Figure supplement 2.** General overview in HS vs NHS.

**Figure supplement 3.** Gene expression analysis in HS vs NHS.

**Figure supplement 4.** Transcript expression in HS vs NHS.

1 hr PSI, we observed a slight separation between the two conditions in the PCA, with PC1 accounting for 13% of the variance and PC2 accounting for 11% (*Figure 4B*, *Figure 4—figure supplement 3A-B*). Similar trends were observed at the transcript level (*Figure 4—figure supplement 4A–B*). We identified three annotated genes including *HSPA1A*, *DNAJB1*, and *HSP90AA1* enriched in the HS condition (*Figure 4C-D*, *Figure 4—figure supplement 3C-D*). We observed similar dynamics in the identified DETs (*Figure 4—figure supplement 4C–D*). These genes were shown to be upregulated under heat shock conditions in previous reports (*Neueder et al., 2014*; *Yonezawa and Bono, 2023*; *Sanchez-Briñas et al., 2023*), demonstrating the efficacy of NanopoReaTA in capturing the expected DEGs early during the sequencing process.

At the 24 hr PSI, we noted a more pronounced separation between the conditions, with PC1 representing 36% of the variance and PC2 representing 23% (*Figure 4E*, *Figure 4—figure supplement 3A*). This clustering trend was consistent throughout the entire sequencing process and was evident in both DGE and DTE analyses (*Figure 4—figure supplements 3A-B and 4A-B*). In total, we identified 43 genes enriched in the HS condition and 7 genes enriched in the NHS condition (*Figure 4F-G*, *Figure 4—figure supplement 3C-D*).

We compared the total differentially expressed genes identified at each time point to evaluate the dynamic changes detected during sequencing. Notably, the three heat shock-associated genes, *HSPA1A*, *DNAJB1*, and *HSP90AA1*, were consistently detected across the entire sequencing process (*Figure 4H*).

Next, we performed Gene Ontology (GO) term analysis using ShinyGO (*Ge et al., 2020*) to examine the biological processes associated with the HS condition. This analysis revealed terms such as 'de novo protein folding,' 'response to unfolded protein', and 'response to heat' among the enriched DEGs in HS, confirming our expected results (*Figure 4I*, *Figure 4—source data 1*). These results demonstrate NanopoReaTA's ability to rapidly identify biologically relevant information even during the early stages of sequencing.

Lastly, we employed DEXSeq and DRIMSeq for differential transcript usage analysis to identify specific transcripts enriched in each condition. We identified the DYNLL1-207 isoform as upregulated in HS compared to NHS (*Figure 4J*). *DYNLL1* is a dynein light chain involved in intracellular transport, apoptosis regulation, and cancerogenesis (*Liu et al., 2024*). Although there is no direct evidence linking *DYNLL1* to the heat shock response, its known roles in stress-related pathways and cellular homeostasis suggest it may have an indirect or previously unexplored involvement in the stress response. NanopoReaTA's ability to detect differential transcript usage offers a valuable tool for isoform-specific research, especially when paired with appropriate experimental designs.

## NanopoReaTA offers rapid quality control assessments for experimental manipulated samples

In our final aim, we sought to highlight the flexibility of NanopoReaTA in an experimental manipulation setup and its applicability beyond human cell culture. To achieve this, we employed *S. cerevisiae* strains harboring gene knockouts or strains transformed with plasmids containing the deleted gene for overexpression. Two distinct experimental setups were designed to assess the reproducibility and detection capabilities of NanopoReaTA.

In the first experimental setup (Yeast setup 1), we utilized *new1Δ::KanMX* yeast strains, where the *NEW1* gene (coding sequence only) was replaced with the KanMX cassette which contains the Kanamycin resistance gene (*KanR*). We used the wild type (WT) strain (BY4741, *MATa, his3Δ1, leu2Δ0, met15Δ0, ura3Δ0*) for comparison with the KO strain. These strains were transformed with either an empty vector with the *HIS3* selection marker (pEV(*HIS3*)) or an overexpression vector built on the same

backbone, for C-terminally FLAG-tagged New1 with the same *HIS3* selection marker (pNew1(*HIS3*); *Figure 5A*, *Figure 5—figure supplement 1A*).

In the second experimental setup (setup 2), we employed the *rkr1Δ::HphMX* strain, where the coding sequence of the *RKR1* gene was replaced with the *HphMX cassette encoding the* Hygromycin B resistance gene (*HygR*). Additionally, we used the double KO strain *jlp2Δ::KanMX, rkr1Δ::HphMX*, where the coding sequence of the *JLP2* gene was replaced with KanMX cassette containing *KanR* (each condition was tested in triplicate). For this setup, these strains were transformed with either an empty vector with the *URA3* selection marker (pEV(*URA3*)) or an overexpression vector for C-terminally HA-tagged Jlp2 with the *URA3* selection marker (pJlp2(*URA3*); *Figure 6A*, *Figure 6—figure supplement 1A*).

All *S. cerevisiae* knockout strains derived from BY4741 were prepared using homologous recombination following standard procedures, and each condition was tested in (biological) triplicates. Furthermore, we employed customized yeast genome annotation files that included *KanR*, Hygromycin resistance gene (*HygR*) and Ampicillin resistance gene (*AmpR*; contained to allow propagation of the shuttle vectors in *E. coli*) transcripts, ensuring the detection of foreign transcripts specific to their corresponding experimental setup, thus adding an extra layer of quality control (see 'Materials and methods'). For the following section, we will focus on describing the detected changes in differentially expressed genes/transcripts; however, general sequencing overviews, as well as detailed analyses for all individual time points are presented in *Figure 5—figure supplements 1–17* (yeast setup 1) and *Figure 6—figure supplements 1–14* (for yeast setup 2). For each experimental setup, we collected data from NanopoReaTA, including general sequencing overviews with the count of detected genes (*Figure 5—figure supplement 1*, *Figure 6—figure supplement 1*), with individual and combined read length distribution (*Figure 5—figure supplements 3, 6*, *Figure 5—figure supplements 9 and 12*, *Figure 5—figure supplement 15* for yeast setup 1; *Figure 6—figure supplements 3, 6*; *Figure 6—figure supplements 9 and 12* for yeast setup 2), gene expression variability (*Figure 5*, *Figure 6—figure supplement 2*), and the timing of tool utilization. We conducted real-time analyses for both DGE and DTE, documenting all associated information at each time point (*Figure 5—source data 1*, and *Figure 6—source data 1*).

## Yeast setup 1

*new1Δ*-pEV(*HIS3*) vs WT-pEV(*HIS3*). For yeast setup 1, we aimed to test NanopoReaTA's capabilities in foreign gene detection since the coding sequence of the *NEW1* gene was replaced via homologous recombination with the KanMX cassette, which contains the *KanR* antibiotic resistance gene. In the comparison of *new1Δ*-pEV(*HIS3*) versus WT-pEV(*HIS3*) at 1 hr PSI, the PCA revealed separation between the samples based on their respective conditions (*Figure 5*, *Figure 5—figure supplements 4 and 5*), as well as clustering in the sample-to-sample distance plot (*Figure 5—figure supplements 4B and 5B*). This clustering pattern persisted until the 24 hr PSI time point (*Figure 5C*, *Figure 5—figure supplements 4A and 5A*). Notably, in every comparison group, we noted one sample that exhibited a slight separation from the condition cluster in the PCA, in contrast to the other two replicates. However, these differences did not raise significant concerns. Nonetheless, the real-time PCA clustering feature of NanopoReaTA could prove valuable when assessing biological replicates.

Upon conducting differential gene and transcript expression analysis at 1 hr PSI, we detected 13 genes enriched and 3 genes depleted in *new1Δ*-pEV(*HIS3*) compared to *WT*-pEV(*HIS3*) (*Figure 5—figure supplement 4C*). These observations were consistent throughout the entire sequencing period, extending up to the 24 hr mark (*Figure 5D-E*, *Figure 5—figure supplement 4C-F*, *Figure 5—source data 1*). These results demonstrate the possibility of detection of foreign transcripts incorporated instead of knockout gene. A previous study conducted RNA-seq between *new1Δ* and WT (*Kasari et al., 2019*); therefore, we overlapped the identified DEGs to examine the commonality between the detected DEGs. Within the overlap, four upregulated genes and 11 downregulated genes, including *HSP12* and *NEW1*, were found to be common between *Kasari et al., 2019* and our investigation (*Figure 5—figure supplement 4G*), despite variations of growth conditions, as well as exact yeast genotypes between this study and *Kasari et al., 2019*. As demonstrated, NanopoReaTA can swiftly identify an experimental knockout, and in instances where the gene is replaced with a foreign gene, it can also detect this alteration effectively given that the foreign gene is incorporated into the genome annotation files.

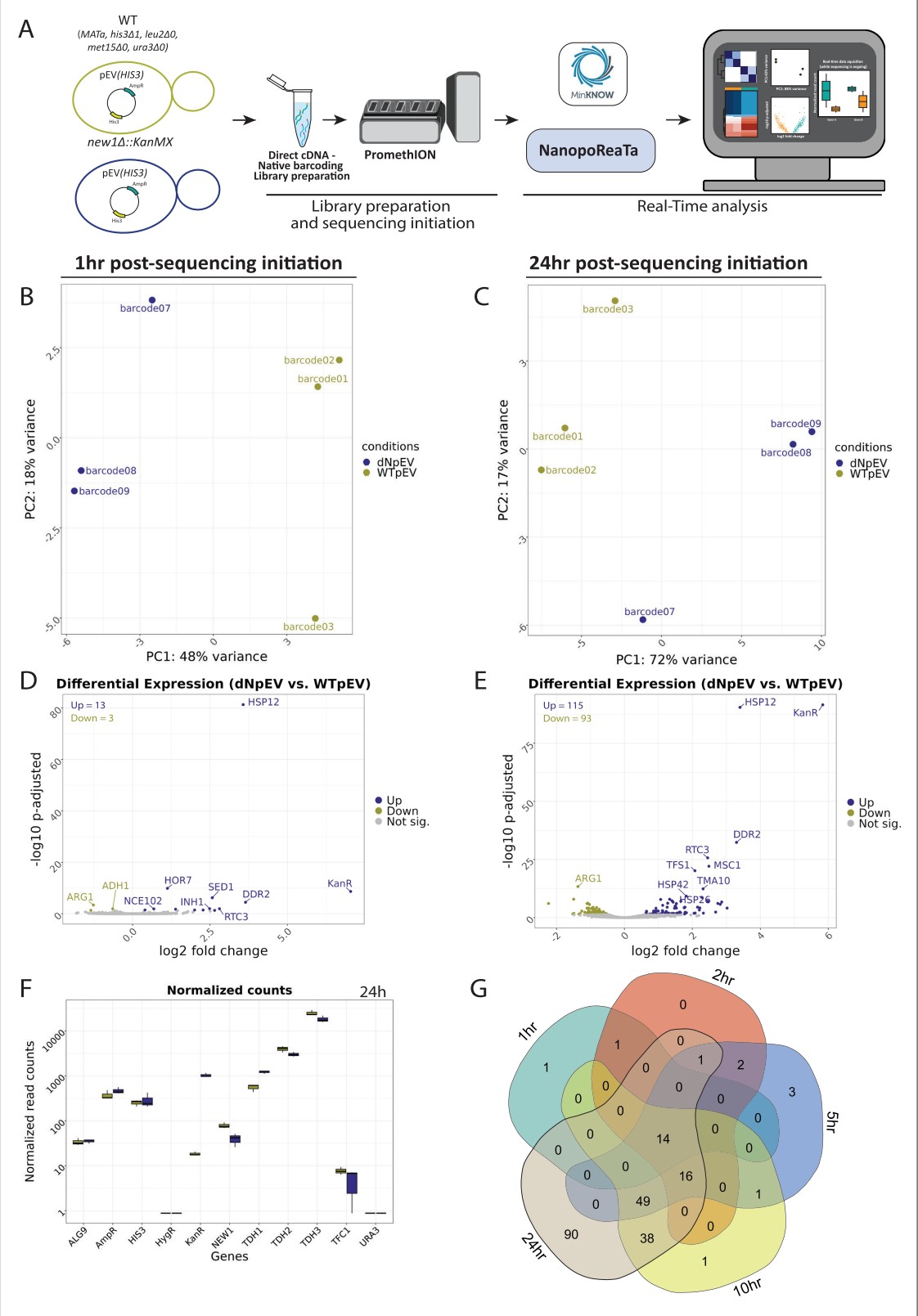

**Figure 5.** Real-time transcriptomic analysis between *new1Δ*-pEV(*HIS3*) vs WT-pEV(*HIS3*) using NanopoReaTA. (**A**) Experimental strategy. Following library preparation (n=3), samples were loaded and sequenced using a PromethION R10 flow cell. NanopoReaTA was activated shortly after sequencing initiation and data were collected 1 hr, 2 hr, 5 hr, 10 hr, and 24 hr post-sequencing initiation. (**B–C**) PCA plot showing the sample separation between *new1Δ*-pEV(*HIS3*) vs WT-pEV(*HIS3*) 1 hr post-sequencing (**B**) and 24 hr post-sequencing (**C**). (**D–E**) Volcano plots showing the differentially expressed

*Figure 5 continued on next page*

*Figure 5 continued*

genes between *new1Δ*-pEV(*HIS3*) vs WT-pEV(*HIS3*) 1 hr post-sequencing initiation (**D**) and 24 hr post-sequencing initiation (**E**). (**F**) Normalized gene count for selected genes 24 hr post-sequencing initiation. Normalized gene counts are visualized for selected genes per condition using boxplots. The median-of-ratios normalization method from DESeq2 was used for normalization. (**G**) Five-way Venn diagram showing the differentially expressed gene overlaps between the distinct collected time points.

The online version of this article includes the following source data and figure supplement(s) for figure 5:

**Source data 1.** General sequencing overview of Yeast setup 1.

**Figure supplement 1.** Sequencing overview yeast setup 1.

**Figure supplement 2.** Real-time transcriptomic analysis in yeast setup 1 samples using NanopoReaTA.

**Figure supplement 3.** General overview in *new1Δ*-pEV(*HIS3*) *vs* WT-pEV(*HIS3*).

**Figure supplement 4.** Gene expression in *new1Δ*-pEV(*HIS3*) *vs* WT-pEV(*HIS3*).

**Figure supplement 5.** Transcript expression *new1Δ-pEV(HIS3) vs WT-pEV(HIS3)*.

**Figure supplement 6.** General overview in WT-pNew1(*HIS3*) *vs* WT-pEV(*HIS3*).

**Figure supplement 7.** Gene expression in WT-pNew1(*HIS3*) *vs* WT-pEV(*HIS3*).

**Figure supplement 8.** Gene expression in WT-pNew1(*HIS3*) *vs* WT-pEV(*HIS3*).

**Figure supplement 9.** General overview in *new1Δ*-pNew1(*HIS3*) *vs* *new1Δ*-pEV(*HIS3*).

**Figure supplement 10.** Gene expression in *new1Δ*-pNew1(*HIS3*) *vs* *new1Δ*-pEV(*HIS3*).

**Figure supplement 11.** Transcript expression in *new1Δ*-pNew1(*HIS3*) *vs* *new1Δ*-pEV(*HIS3*).

**Figure supplement 12.** General overview in *new1Δ*-pNew1(*HIS3*) *vs* WT-pEV(*HIS3*).

**Figure supplement 13.** Gene expression in *new1Δ*-pNew1(*HIS3*) *vs* WT-pEV(*HIS3*).

**Figure supplement 14.** Transcript expression in *new1Δ*-pNew1(*HIS3*) *vs* WT-pEV(*HIS3*).

**Figure supplement 15.** General overview in *new1Δ*-pNew1(*HIS3*) *vs* WT-pNew1(*HIS3*).

**Figure supplement 16.** Gene expression in *new1Δ*-pNew1(*HIS3*) *vs* WT-pNew1(*HIS3*).

**Figure supplement 17.** Transcript expression in *new1Δ*-pNew1*(HIS3) vs* WT-pNew1*(HIS3)*.

*WT-pNew1(HIS3) vs WT-pEV(HIS3)*. Next, we assessed NanopoReaTA's capability to identify the expressed content of the transformed plasmids by contrasting the WT strain transformed with pNew1(*HIS3*) against the WT strain transformed with pEV(*HIS3*). In the comparison of WT pNew1(*HIS3*) versus WT-pEV(*HIS3*) at 1 hr PSI, the PCA effectively distinguished the samples based on their respective conditions (***Figure 5—figure supplement 2H***, ***Figure 5—figure supplement 7A***). However, the sample-to-sample distance plot did not uncover significant differences between the replicates (***Figure 5—figure supplement 7B***). The distinct clustering observed in PCA was consistently maintained throughout the entire sequencing process until the 24 hr PSI mark (***Figure 5—figure supplements 2H-I and 7A***). The sample-to-sample distance plot indicated greater similarities between the samples, which is anticipated given the comparison involves similar WT strains harboring either pNew1(*HIS3*) or pEV(*HIS3*). Notably, *NEW1* was the sole differentially expressed gene (***Figure 5—figure supplement 2J***) and transcript (***Figure 5—figure supplement 8C–D***) identified at the 1 hr PSI mark in the WT strain supplemented with pNew1(His3), in comparison to the WT strain *transformed* with pEV(*HIS3*). This difference was maintained throughout the whole sequencing (***Figure 5—figure supplement 2K–M***, ***Figure 5—source data 1***). Using the 'Gene-wise' feature in NanopoReaTA, we tracked, in real-time, the normalized read counts of various genes of interest, such as *NEW1*, *KanR*, *AmpR*, and *HIS3*, along with housekeeping genes such as *ALG9 and TFC1* (***Teste et al., 2009***), and the commonly used versions of yeast Gapdh-encoding genes *TDH1*, *TDH2*, and *TDH3* (***Figure 5—figure supplement 2L***, ***Figure 5—figure supplement 7E***). This setup demonstrates an overexpression experiment, showcasing NanopoReaTA's capability to swiftly detect the overexpressed gene.

*new1Δ-pNew1(HIS3) vs new1Δ-pEV(HIS3)*. Next, we compared *new1Δ* strains harboring either pNew1(*HIS3*) or pEV(*HIS3*) at 1 hr PSI. As in WT, the PCA successfully differentiated the samples according to their respective conditions (***Figure 5—figure supplement 2N***, ***Figure 5—figure supplement 8A***). Nevertheless, the sample-to-sample distance plot did not reveal notable differences between the replicates (***Figure 5—figure supplement 8B***). The distinct clustering observed in PCA remained consistent throughout the entire sequencing process, until the 24 hr PSI point (***Figure 5—figure supplement 2O***, ***Figure 5—figure supplement 8A***). NEW1 emerged as the only gene and

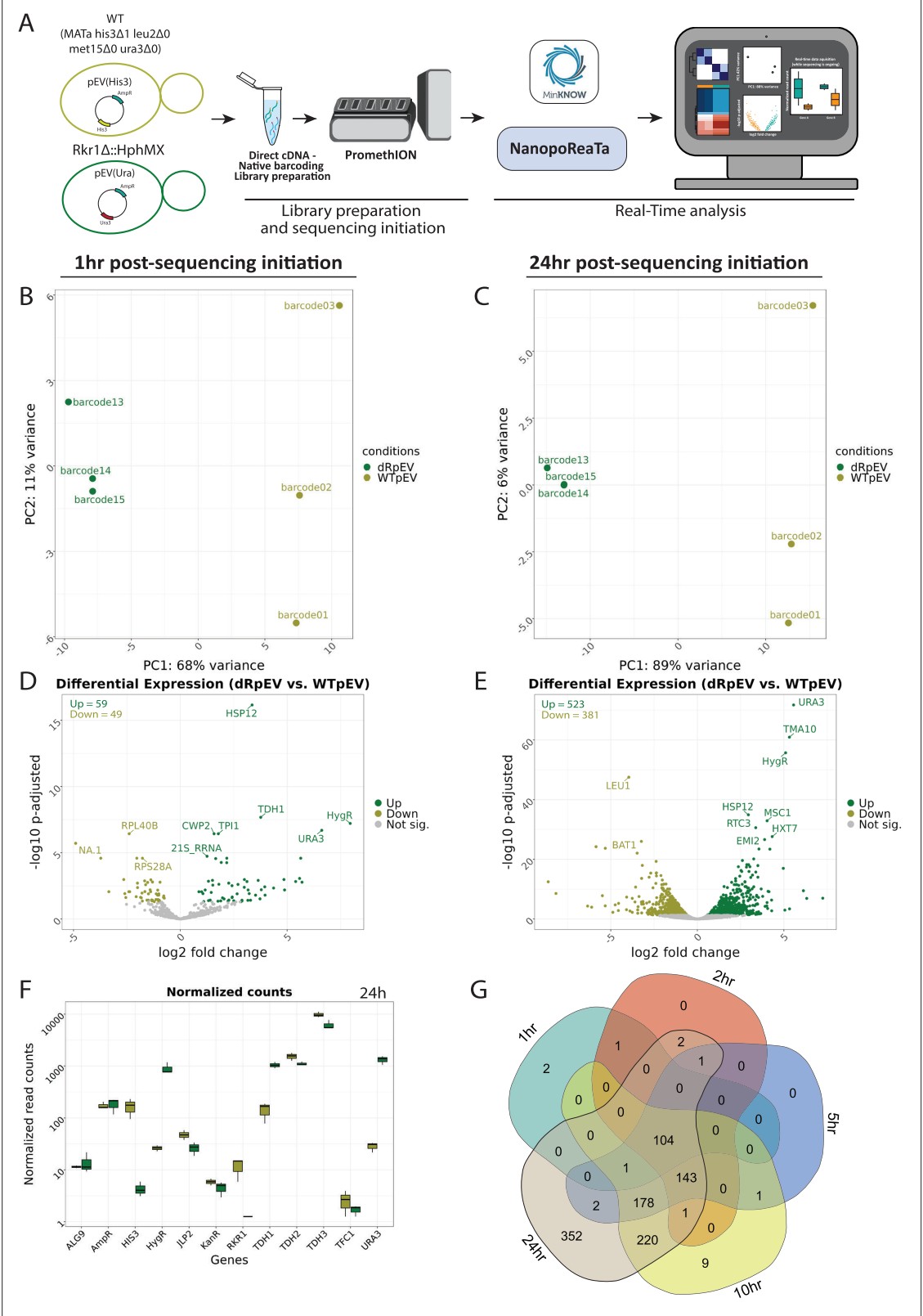

**Figure 6.** Real-time transcriptomic analysis between *rkr1Δ*-pEV(*URA3*) vs WT-pEV(*HIS3*) using NanopoReaTA. (**A**) Experimental strategy. Following library preparation (n=3), samples were loaded and sequenced using a PromethION R10 flow cell. NanopoReaTA was activated shortly after sequencing initiation and data was collected 1 hr, 2 hr, 5 hr, 10 hr, and 24 hr post-sequencing initiation. (**B–C**) PCA plot showing the sample separation between *rkr1Δ*-pEV(*URA3*) vs WT-pEV(*HIS3*) 1 hr post-sequencing initiation (**B**) and 24 hr post-sequencing initiation (**C**). (**D–E**) Volcano plots showing the

*Figure 6 continued on next page*

Butto, Pastore *et al*. eLife 2024;13:RP98768. DOI: https://doi.org/10.7554/eLife.98768 15 of 33

*Figure 6 continued*

differentially expressed genes between *rkr1Δ*-pEV(*URA3*) vs WT-pEV(*HIS3*) 1 hr post-sequencing initiation (**D**) and 24 hr post-sequencing initiation (**E**). (**F**) Normalized gene count for selected genes 24 hr post sequencing initiation. Normalized gene counts are visualized for selected genes per condition using boxplots. The median-of-ratios normalization method from DESeq2 was used for normalization. (**G**) Five-way Venn diagram showing the differentially expressed gene overlaps between the distinct collected time points.

The online version of this article includes the following source data and figure supplement(s) for figure 6:

**Figure supplement 15.** IGV coverage snapshots of selected gene loci in yeast setup 1 and 2.

transcript enriched in pNew1(*HIS3*) as compared to pEV(*HIS3*) at 1 hr PSI (*Figure 5—figure supplement 2P*, *Figure 5—figure supplement 8C–D*). This observed difference persisted throughout the entire sequencing process (*Figure 5—figure supplement 2Q*, *Figure 5—figure supplement 8C–F*, *Figure 5—source data 1*). Interestingly, as sequencing was going through, the number of DEG/T increased up until the 24 hr PSI mark where 13 genes were enriched and 2 were depleted in *new1Δ*-pNew1(*HIS3*) compared to *new1Δ*-pEV(*HIS3*) (*Figure 5—figure supplement 2Q–S*). This experimental configuration exemplifies a rescue experiment, highlighting NanopoReaTA's ability to promptly detect the overexpressed gene.

*new1Δ*-pNew1(*HIS3*) vs WT-pEV(*HIS3*). When comparing *new1Δ*-pNew1(*HIS3*) versus WT-pEV(*HIS3*) at 1 hr PSI, the PCA indicated separation between the samples based on their respective conditions (*Figure 5—figure supplements 13A and 14A*); however, the clustering in the sample-to-sample distance plot appeared inconsistent, likely due to the rescue of New1 in the *new1Δ* mutant strain (*Figure 5—figure supplements 13B and 14B*). Remarkably, we consistently detected both *KanR* and *NEW1* as enriched in *new1Δ*-pNew1(*HIS3*) compared to WT-pEV(*HIS3*) throughout the entire sequencing process, and both as differentially expressed genes and transcripts (*Figure 5—figure supplements 13C-F and 14C-E*, *Figure 5—source data 1*). These observations highlight NanopoReaTA's rapid detection capabilities in an experimental setup where the deleted gene is rescued by the overexpression plasmid compared to WT with an empty vector.

*new1Δ*-pNew1(*HIS3*) versus WT-pNew1(*HIS3*). In the comparison of *new1Δ*-pNew1(*HIS3*) versus WT-pNew1(*HIS3*) at 1 hr PSI, the PCA exhibited separation between the samples based on their respective conditions (*Figure 5—figure supplements 2T*; *Figure 5—figure supplements 16A and 17A*), accompanied by clustering in the sample-to-sample distance plot (*Figure 5—figure supplements 16B and 17B*). This clustering pattern persisted until the 24 hr PSI time point (*Figure 5—figure supplements 1V*; *Figure 5—figure supplements 16A and 17A*). *KanR*, *DDR2*, and *HSP12* were identified as enriched genes in *new1Δ*-pNew1(*HIS3*) compared to WT-pNew1(*HIS3*) at 1 hr time point, appearing as both differentially expressed genes and transcripts (*Figure 5—figure supplements 2U*; *Figure 5—figure supplements 16C-F and 17C-E*). At 24 hr PSI, only 29 genes were enriched and 2 were depleted in *new1Δ*-pNew1(*HIS3*) compared to WT-pNew1(*HIS3*), implying a potential

transcriptional overcompensation facilitated by the transformed plasmid encoding *NEW1* (*Figure 5—figure supplements 2W-Y*; *Figure 5—figure supplements 16C-F and 17C-E*, *Figure 5—source data 1*).

It is important to highlight that using the 'gene-wise' utility, we identified discrepancies in the reads associated with each condition. For instance, in the *new1Δ* condition, where the *NEW1* gene (coding sequence only) has been replaced with the *KanMX* cassette containing the *KanR* gene, some *NEW1* transcripts still aligned and were quantified. Additionally, we observed the presence of condition-specific transcripts (e.g. *KanR*, expected only in *new1Δ* mutants) in WT conditions, although in low quantities. A detailed discussion of these discrepancies is provided in Appendix 3. These findings highlight the ability of NanopoReaTA to offer valuable quality control insights that could reveal experimental flaws, such as contaminations, which could then be rapidly addressed and rectified.

## Yeast setup 2

*rkr1Δ*-pEV(*URA3*) versus WT-pEV(*HIS3*). First, we assessed the KO strains with the WT strain, utilizing WT-pEV(*HIS3*), and detecting the distinct selection genes present in each condition. Initially, we conducted a comparison between *rkr1Δ*-pEV(*URA3*) and WT-pEV(*HIS3*) (*Figure 6A*, *Figure 6—figure supplement 2B-G*). At 1 hr PSI, a notable separation between the conditions was evident in both the PCA and sample-to-sample distance plot (*Figure 6B*, *Figure 6—figure supplements 4A-B and 5A-B*). This distinction persisted consistently until the 24 h PSI time point (*Figure 6C*, *Figure 6—figure supplements 4A-B and 5A-B*). At 1 hr PSI, 59 genes were enriched and 49 were detected as depleted in *rkr1Δ*-pEV(*URA3*) as compared to WT-pEV(*HIS3*). As expected, *URA3* and *HygR* were detected as enriched in *rkr1Δ*-pEV(*URA3*) (*Figure 6D*, *Figure 6—figure supplements 4C-D and 5C-D*). At this stage, *HIS3* was not identified in the DEG analysis, but rather in the DET analysis. At the 24 hr PSI mark, 523 genes were enriched (including *URA3* and *HygR*) and 381 were depleted (including *HIS3*) in *rkr1Δ*-pEV (*URA3*) as compared to WT-pEV (*HIS3*) (*Figure 6E-F*, *Figure 6—figure supplements 4C-F and 5C-E*). Interestingly, 104 differentially expressed genes were consistently identified from the 1 hr mark until the final 24 hr mark (*Figure 6G*, *Figure 6—source data 1*). This experimental configuration demonstrates the detection of the knockout of an individual gene, the detection of *HygR*, *URA3*, and *HIS3* selection genes as well as a large number of additional DEGs/DETs that could validate the mechanistic function of the mutant investigated, as well as differences between yeast grown in different culturing conditions.

*rkr1Δ*-pJlp2(*URA3*) *versus rkr1Δ*-pEV(*URA3*). Next, we tested the comparison of *rkr1Δ*-pJlp2(*URA3*) compared to *rkr1Δ*-pEV(*URA3*) for the detection of *JLP2* overexpression. One hour after the initiation of sequencing, PCA successfully differentiated the samples based on their respective conditions, with a slight separation observed in one replicate (barcode 16- *rkr1Δ*-pJlp2(*URA3*)) (*Figure 6—figure supplements 2H*; *Figure 6—figure supplements 7A and 8A*). The sample-to-sample distance plot did not uncover significant differences between the replicates (*Figure 6—figure supplements 7B and 8B*). The distinct clustering observed in PCA was consistently maintained throughout the entire sequencing process (*Figure 6—figure supplements 7A and 8A*) until the 24 hr PSI mark (*Figure 6—figure supplements 2I*; *Figure 6—figure supplements 7A and 8A*). Notably, two genes/transcripts including *JLP2* and *RPL15A* were differentially expressed from 1 hr PSI mark until 24 hr PSI mark (*Figure 6—figure supplements 2J-M*; *Figure 6—figure supplements 7C-D and 8C-D*, *Figure 6—source data 1*). Thus, NanopoReaTA was able to detect the overexpressed gene from the plasmid: *rkr1Δ, jlp2Δ*-pJlp2(*URA3*) versus *rkr1Δ, jlp2Δ*-pEV(*URA3*). Similarly, we tested *rkr1Δ, jlp2Δ*-pJlp2(*URA3*) compared to *rkr1Δ, jlp2Δ*-pEV(*URA3*) for the detection of *JLP2* overexpression. When comparing *rkr1Δ, jlp2Δ*-pJlp2(*URA3*) versus *rkr1Δ, jlp2Δ*-pEV(*URA3*) at 1 hr PSI, the PCA showed separation between the samples based on their respective conditions (*Figure 6—figure supplements 2N*; *Figure 6—figure supplements 10A and 11A*); similarly, the clustering in the sample-to-sample distance plot appeared inconsistent (*Figure 6—figure supplements 10B and 11*). The clustering observed in PCA remained consistent throughout the entire sequencing process until the 24 hr PSI time point (*Figure 6—figure supplements 2O*; *Figure 6—figure supplements 10A and 11A*). Remarkably, we detected only *Jlp2* as enriched in *rkr1Δ, jlp2Δ*-pJlp2(*URA3*) compared to *rkr1Δjlp2Δ*-pEV(*URA3*) throughout the entire sequencing process, in DEG and DET analyses (*Figure 6—figure supplements 2P-S*; *Figure 6—figure supplements 10C-F and 11C-D*, *Figure 6—source data 1*). At the 24 hr PSI mark, we observed enrichment of *JLP2* and depletion of 14 genes in *rkr1Δ, jlp2Δ*-pJlp2(*URA3*)

compared to *rkr1Δ, jlp2Δ*-pEV(*URA3*). These observations highlight the swift and accurate detection capabilities of NanopoReaTA in an experimental setup where similar strains are compared, and only one is transformed with an overexpression vector, thereby illustrating the rescue by the expressed gene.

*rkr1Δ, jlp2Δ*-pEV(*URA3*) versus WT-pEV(*HIS3*). Lastly, we compared *rkr1Δ, jlp2Δ*-pEV(*URA3*) to WT-pEV(*HIS3*). At 1 hr PSI, noticeable separation between conditions was evident in both PCA and sample-to-sample distance plots (***Figure 6—figure supplements 2T***; ***Figure 6—figure supplements 13A-B and 14A-B***). This distinction persisted consistently until the 24 hr PSI time point (***Figure 6—figure supplements 2V***; ***Figure 6—figure supplements 13A-B and 14A-B***). At 1 hr PSI, 69 genes were enriched, and 34 were depleted in *rkr1Δ*-pEV(*URA3*) compared to WT-pEV(*HIS3*). As expected, *URA3* and *HygR* were detected as enriched in *rkr1Δ*-pEV(*URA3*) (***Figure 6—figure supplement 2U***; ***Figure 6—figure supplements 13C-D and 14C-D***). Similar to the previous observation, *HIS3* was not identified in the DEG analysis but rather in the DET analysis. By the 24 hr PSI mark, 612 genes were enriched (including *URA3* and *HygR*), and 424 were depleted (including *HIS3*) in *rkr1Δ*-pEV(*URA3*) compared to WT-pEV(*HIS3*) (***Figure 6—figure supplements 2W-Y***; ***Figure 6—figure supplements 13C-E and 14C-D***).

A total of 101 differentially expressed genes were consistently identified from the 1 hr mark until the final 24 hr mark (***Figure 6—source data 1***). This experimental setup effectively detected the double knockout, as well as the expression of *HygR*, *KanR*, *URA3*, and *HIS3* in their corresponding experimental conditions.

Similar to yeast experimental setup 1, we observed unexpected findings in the reads associated with each condition using the 'gene-wise' utility. For example, JLP2 expression was detected in *rkr1Δjlp2Δ*-pEV(*URA3*) in low quantities, and *HIS3* was observed in strains where the selection plasmid should not contain the *HIS3* selection marker. A comprehensive discussion of these observations is presented in **Appendix 3**. Nonetheless, NanopoReaTA can rapidly detect experiment-specific transcripts associated with the experimental condition. This application can be utilized to quickly identify knockout, knockdown, or overexpression experiments and to quantify foreign transcripts that are not naturally present in the species' genome.

## Discussion

We presented a proof-of-concept application use of NanopoReaTA demonstrating its rapid detection capabilities of pairwise transcriptomic changes and for the first time, real-time dynamics of long read RNA-seq throughout the sequencing process. NanopoReaTA can be used as a multi-species transcriptomic detection tool revealing its broad utility. The tool requires well-annotated genomes including genome sequence (FASTA files), annotated transcripts (FASTA files), gene annotation (GTF files), and gene coordinates throughout the genome (BED files). Additionally, NanopoReaTA works in combination with MinION/GridION flow cells; however, due to their reduced throughput compared to PromethION flow cells, achieving statistically meaningful results (e.g. larger number of DEGs) may be limited or take longer.

The straightforward utilization of NanopoReaTA, coupled with an intuitive graphical user interface (GUI), facilitates its smooth integration into daily experimental setups for quality checks in transcriptomic data analysis. The tool swiftly identifies transcriptomic differences between distinct cell types, compartment-enriched transcripts, or genetically manipulated cells, even within the first hour post-sequencing initiation. It is highly probable that these early detected changes represent the most significant transcripts, present or highly expressed in one condition versus absent or lowly expressed in another condition. These noteworthy early alterations persist throughout the entire sequencing process until its completion. As sequencing progresses and more reads are acquired, there is an increase in the number of detected genes, as well as genes and transcripts detected as differentially expressed (DEGs and DETs). It is important to note that these DEGs and DETs may undergo changes over the sequencing process as the data is normalized to the total read counts within the compared conditions (***Evans et al., 2018***). We incorporated into NanopoReaTA both differential gene/transcript expression analyses, performed by DESeq2 (***Love et al., 2014***), and quantification of genes and transcripts was performed by featureCounts (***Liao et al., 2014***) and Salmon (***Patro et al., 2017***), respectively. It is acknowledged that utilizing different analysis tools may lead to detection of varying numbers and tool-specific differentially expressed genes or transcripts (***Thawng and Smith, 2023***).

Therefore, by offering both analyses, our intention is to provide orthogonal methodologies, ensuring that the most significant outcomes are consistently identified across different methods. Moreover, given the capability of capturing complete transcripts with long-read sequencing, we integrated a 'differential transcript usage' application performed by DEXseq (*Anders et al., 2012*) and DRIMSeq (*Nowicka and Robinson, 2016*). These applications are dedicated to the analysis and quantification of different isoforms per selected gene. This utility proves beneficial in uncovering or determining the predominant isoform used between two conditions, and utilizing it more frequently could unveil novel biological insights.

NanopoReaTA provides multi-layer quality control of several distinct experimental setups. On the first layer, NanopoReaTA can provide information regarding the number of genes identified, both per sample and per condition, as well as the changes in gene composition detected in each iteration compared to the previous one. When no additional genes are detected, the 'Gene expression variability' lines reach a plateau, and the sequencing can be practically terminated (depending on the desired read depth). Such quality control provides a cost-efficient strategy when coupled with the Nanopore-seq washable flow cell that can be reused for separate experimental setups. Moreover, this analysis provides relevant biological insights into the number of genes expressed under specific conditions, a factor that may vary across different cell types or distinct experimental conditions. Another level of quality control can be applied when comparing distinct cell types or strains, where several cell-type/strain specific gene markers can be examined. Using the 'Gene-wise' utility, these marker genes could be monitored in real-time, providing quality control for the cell-type/strain-specific purity as compared to distinct cell type. Combined with this, a third layer of quality control is featured while performing differential gene/transcript expression with the visualization of the PCA. Such analysis could reveal rapidly the transcriptional differences between distinct cell types by monitoring the increased PC variance throughout sequencing. Ideally, similar samples (e.g. technical/biological replicates) would cluster together whereas distinct samples (e.g. distinct conditions/cell types) will cluster separately. Similarly, such analyses could also reveal inter-sample variability between similar biological replicates, providing information about their transcriptional states (similarity or dissimilarities) and thus the reliability of the results. Lastly, NanopoReaTA could analyze foreign expressed genes using modified genome annotations that had incorporated gene sequences which are not naturally present in the species' genome. This was demonstrated in the yeast strains experimental setup with the detection and quantification of foreign genes such as *KanR*, *AmpR,* and *HygR*, providing confirmation of the incorporated mutation or transformation efficiency of the foreign vectors. Such a utility could have a major value when a gene of interest is replaced with a foreign gene (e.g. an antibiotic resistance gene) or when introducing foreign vectors harboring specific selection genes. In practice, NanopoReaTA could also be used to detect fusion-protein transcripts as well as monitor transcription efficiency from specific promoters by quantifying the expressed transcripts. On top of these multi-layered quality control detection capabilities, NanopoReaTA performs long-read RNA-seq data analyses in parallel to ongoing sequencing, providing valuable preliminary results of the experimental setup. In case all the QC criteria are fulfilled, the sequencing can be maintained until reaching the desired sequencing depth.

NanopoReaTA's usefulness in academic settings extends to reducing sequencing costs and enhancing sample quality checks prior to sequencing. However, the potential impact of real-time analysis tools in clinical settings is possibly even more far-reaching. For instance, *Gorzynski et al., 2022* have introduced an efficient framework for whole genome sequencing, setting a world record in the sequencing and analysis of whole genomes. Not only is this approach technically impressive, but it also enables rapid genetic diagnosis, ultimately improving clinical diagnoses and reducing associated costs (*Gorzynski et al., 2022*). Additional possibilities may include employing rapid transcriptomic analyses to identify pathogen-specific transcripts or detection of disease-associated transcripts or transcript isoforms (e.g. detection of aberrant BRCA transcripts). The integration of real-time analysis tools like NanopoReaTA could revolutionize clinical applications as a diagnostic tool, especially when considering transcriptomic data. In conclusion, NanopoReaTA stands out as a valuable tool applicable

in both academic and clinical settings, offering cost-effective quality checks for specific experimental conditions while simultaneously providing valuable data through the execution of long-read RNA-seq.

# Materials and methods

## Key resources table

| Reagent type (species) or resource | Designation | Source or reference | Identifiers | Additional information |
|---|---|---|---|---|
| Cell line (*Homo sapiens*) | HEK293 | ATCC | CRL-1573; RRID:CVCL_0045 | |
| Cell line (*H. sapiens*) | HeLa | ATCC | CCL-2; RRID:CVCL_0030 | |
| Strain (*Saccharomyces cerevisiae*) | BY4741 | *Brachmann et al., 1998* | https://doi.org/10.1002/(SICI)1097-0061(19980130)14:2<115::AID-YEA204>3.0.CO;2–2 Available as well from: Euroscarf: Y00000; ATCC: 4040002 | Genotype: MATa his3Δ1 leu2Δ0 met15Δ0 ura3Δ0 |
| Strain (*S. cerevisiae*) | WTpEV | This study | | Genotype: BY4741+pEV(HIS3); available from M.L. Winz |
| Strain (*S. cerevisiae*) | WTpN | This study | | Genotype: BY4741+pNew1-Flag(HIS3); available from M.L. Winz |
| Strain (*S. cerevisiae*) | dNpEV | *Müller et al., 2026* | https://doi.org/10.1093/nar/gkag047 | Genotype: new1Δ::KanMX +pEV(HIS3); available from M.L. Winz |
| Strain (*S. cerevisiae*) | dNpN | *Müller et al., 2026* | https://doi.org/10.1093/nar/gkag047 | Genotype: new1Δ::KanMX +pNew1-Flag(HIS3); available from M.L. Winz |
| Strain (*S. cerevisiae*) | dRpEV | This study | | Genotype: rkr1Δ::HphMX +pEV(URA3); available from M.L. Winz |
| Strain (*S. cerevisiae*) | dRpJ | This study | | Genotype: rkr1Δ::HphMX +pJLP2 HA(URA3); available from M.L. Winz |
| Strain (*S. cerevisiae*) | dRJpEV | This study | | Genotype: jlp2Δ::KanMX rkr1Δ::HphMX +pEV(URA3); available from M.L. Winz |
| Strain (*S. cerevisiae*) | dRJpJ | *Iyer et al., 2025* | https://doi.org/10.1101/2025.09.04.673968 | Genotype: jlp2Δ::KanMX rkr1Δ::HphMX +pJLP2 HA(URA3); available from M.L. Winz; Note that RKR1 is designated by its alias LTN1 in the cited reference. |
| Recombinant DNA reagent | pEV(HIS3) | *Müller et al., 2026* | https://doi.org/10.1093/nar/gkag047 | Shuttle vector plasmid: empty vector, HIS3 selection marker, available from M.L. Winz |
| Recombinant DNA reagent | pNew1(HIS3) | *Müller et al., 2026* | https://doi.org/10.1093/nar/gkag047 | Shuttle vector plasmid: encoding C-terminal FLAG-tagged New1, HIS3 selection marker, available from M.L. Winz |
| Recombinant DNA reagent | pEV(URA3) | This study | N/A | Shuttle vector plasmid: empty vector, URA3 selection marker, available from M.L. Winz |
| Recombinant DNA reagent | pJlp2(URA3) | *Iyer et al., 2025* | https://doi.org/10.1101/2025.09.04.673968 | Shuttle vector plasmid: encoding C-terminal HA-tagged Jlp2, URA3 selection cassette, available from M.L. Winz |
| Chemical compound, drug | TRIzol | Thermo Fisher Scientific | Cat#: 15596018 | |
| Chemical compound, drug | Chloroform | Roth | Cat#: 7331.2 | |
| Chemical compound, drug | Isopropanol | Roth | Cat#: 6752.2 | |
| Chemical compound, drug | Chloroform:isoamyl alcohol (24:1) | Roth | Cat#: X984.1 | |

*Continued on next page*

*Continued*

| Reagent type (species) or resource | Designation | Source or reference | Identifiers | Additional information |
|---|---|---|---|---|
| Chemical compound, drug | Ethanol | Roth | Cat#: 9065.3 | |
| Commercial assay or kit | Ribominus Eukaryote Kit for RNA-seq | Thermo Fisher / Ambion | Cat#: A10837-08 | |
| Commercial assay or kit | Maxima H Minus Double-Stranded cDNA Synthesis Kit | Thermo Scientific | Cat#: K2561 | |
| Commercial assay or kit | RNaseOUT | Invitrogen | Cat#:10777019 | |
| Commercial assay or kit | NEBNext Ultra II End Repair / dA-Tailing Module | New England Biolabs | Cat#: E7546 | |
| Commercial assay or kit | Native Barcoding Expansion 1–12 | Oxford Nanopore Technologies | Cat#: EXP-NBD104 | |
| Commercial assay or kit | RNase I | Thermo Fisher Scientific | Cat#: EN0601 | |
| Commercial assay or kit | Quick T4 DNA Ligase | New England Biolabs Inc. | Cat#: E6056 | |
| Commercial assay or kit | AMPure XP beads | Beckman Coulter | Cat#: A63881 | |
| Commercial assay or kit | Qubit Fluorometric Quantitation | Thermo Fisher Scientific | Cat# Q33238 | |
| Commercial assay or kit | Flow Cell Wash Kit | Oxford Nanopore Technologies | Cat#: EXP-WSH003 | |
| Commercial assay or kit | PromethION Flow cell (R10.4.1) | Oxford Nanopore Technologies | Cat#: FLO-PRO114M | |
| Commercial assay or kit | MinION Flow Cell (R9.4.1) | Oxford Nanopore Technologies | Cat#: FLO-MIN106 | |
| Software, algorithm | Guppy basecaller | Oxford Nanopore Technologies | v3.6.1 | Used in high-accuracy (hac) mode for PromethION and super-accuracy mode for MinION |
| Software, algorithm | NanopoReaTA | *Wierczeiko et al., 2023* | https://doi.org/10.1093/bioinformatics/btad492 | Real-time transcriptomic analysis pipeline; All associated software/algorithm is reported in this publication. See more information in https://github.com/AnWiercze/NanopoReaTA (*Wierczeiko, 2024*) |

## Cell culture

For HEK293 and HeLa transcriptional comparison, cells were cultured and maintained in an incubator at 37 °C and 5% $CO_2$. HEK293 and HeLa cells were cultured in Dulbecco's modified Eagle medium (DMEM) supplemented with 10% FBS, 1% penicillin-streptomycin, and 1% L-glutamine. Once the cells were confluent, the medium was removed and cells were washed once with 1 mL DPBS. The cells were resuspended with 0.5 mL Trizol and collected for Trizol RNA isolation.

## Yeast strain growth

*S. cerevisiae* knockout strains derived from BY4741 were prepared by homologous recombination using standard procedures. Genotypes and culture media used for the respective strains are given in Key Resources Table. For preparation of *S. cerevisiae* RNA, 3 mL of the respective media were inoculated with a single colony of the respective strain and grown overnight in an orbital shaker (30 °C, 220 rpm). 25 mL of the same media were then inoculated with the respective overnight culture to an $OD_{600}$ of 0.2 and cultured at 30 °C, 220 rpm until an $OD_{600}$ of 0.8–1.0 (log-phase). Cells were harvested by centrifugation at 4 °C, pellets were washed twice with Milli Q water, resuspended in Trizol, and snap-frozen in liquid nitrogen.

Plasmids and media used: pEV(*HIS3*) – Empty vector with *HIS3* selection marker. pNew1(*HIS3*) – Overexpression vector for C-terminally FLAG-tagged New1 with *HIS3* selection marker. pEV(*URA3*) - Empty vector with *URA3* selection marker. pJlp2(*URA3*) - Overexpression vector for C-terminally HA-tagged Jlp2 with *URA3* selection marker.

HIS(-) media: 20 g/L glucose, 6.9 g/L Yeast Nitrogen Base without amino acids (Formedium), 1.4 g/L yeast synthetic complete drop-out medium supplements (Formedium), 76 mg/L of each: L-Tryptophan (Roth), L-Leucine (Roth), and Uracil (Formedium).

URA(-) media: 20 g/L glucose, 6.9 g/L Yeast Nitrogen Base without amino acids, 770 mg/l CSM, Single Drop-Out -Ura (Formedium).

## Heat-shock experiments

HEK293 (1 Mio. cells) were plated on petri dishes (Greiner, diameter 10 cm, REF: 664160). After 48 hr, plates were sealed with Parafilm and submerged in a water bath. Heat shock treatment was 42 °C for 45 min. Plates were put back at 37 °C to let the cells recover from heat shock. Non-induced cells were maintained at 37 °C.

## RNA isolation

### HEK 293 and HeLa

For RNA isolation, following 5 min incubation in RT, 100 µL of chloroform was added. Samples were vortexed and incubated 2 min at RT. Samples were centrifuged at 13,000 x *g*, 4 °C for 10 min and the upper aqueous phase was transferred into a new tube. Next, 250 µL of isopropanol was added and incubated for 15 min at RT for RNA precipitation. The samples were then centrifuged at 13,000 – 15,000 x *g* at 4 °C for 30 min and the supernatant was discarded. The RNA pellet was washed with cold 75% cold EtOH (stored at –20 °C) and centrifuged again at 13,000 – 15,000 x *g* at 4 °C for 30 min. The supernatant was discarded, and the pellet was air-dried. The RNA pellet was resuspended with nuclease-free water, and concentration was measured using nanodrop. RNA samples with A260/280 and A260/230 absorbance values >1.9 were taken for library preparation.

### Yeast

Cells in Trizol were thawed on ice and disrupted by bead-beating with zirconia/glass beads (0.5 mm) and vortexing 10 times in cycles of 30 s vortexing at 3000 rpm, intermittent with at least 30 s chilling on ice. Following this, 150 µL of Chloroform/Isoamyl alcohol (24:1 V/V) were added per 750 µL of Trizol and vortexed. After centrifugation (10 min, 14,000 rpm, 4 °C) and an optional second extraction of the aqueous phase with Chloroform/Isoamyl alcohol (24:1 V/V) and water-saturated phenol (pH 4,5–5), the aqueous phase was mixed with sodium acetate pH 5.2 (to at least 0.15 M) and RNA was precipitated by addition of 2-propanol and centrifugation (20–30 min, 14,000 rpm, 4 °C). The pellet was washed twice with ice-cold 75% ethanol, briefly dried, and dissolved in Milli Q water. RNA concentrations were measured by nanodrop and RNA samples with A260/280 and A260/230 absorbance values >1.9 were taken for library preparation.

## Selective purification of ribosomal-depleted (Ribominus) and ribosomal-enriched (Riboplus) transcripts

Selective purification of distinct RNA populations was performed using the Ribominus Eukaryote kit for RNA-seq (#Ambion, A10837-08) according to the manufacturer's 'standard protocol' instructions, with slight modifications for specific rRNA isolation. For the procedure, 5 µg of RNA in 5 µL nuclease-free water was subjected to hybridization with 100 µL of Hybridization Buffer and 10 µL of Ribominus Probe (15 pmol/µL) at 70–75°C for 5 min, followed by an additional 30-min incubation at 37 °C. The Ribominus Magnetic beads were prepared according to the manufacturer's instructions. The RNA/probe mixture was then combined with Ribominus Magnetic beads and incubated at 37 °C for 15 min. Subsequently, magnetic separation was employed to pellet the rRNA-probe complex, and the supernatant, containing ribo-depleted RNA, was collected. The remaining beads underwent a similar process for rRNA isolation using nuclease-free water, and the resulting Ribominus RNA was added to the previous supernatant. To isolate the RNA from the Ribominus supernatant, the sample underwent ethanol precipitation according to the manufacturer's instructions. The pooled bead samples (containing the rRNA) were further processed with Trizol RNA isolation to complete the purification of

the ribosomal-enriched samples (Riboplus). Qualitative analysis of the material was performed after rRNA depletion and enrichment. 1 µg of Total RNA from HEK293 and Riboplus and 150 ng of Ribominus were assessed on 1% TBE agarose gel stained with ethidium bromide.

## Direct cDNA-native barcoding nanopore library preparation and sequencing

Double-stranded cDNA synthesis was carried out using the Maxima H Minus Double-Stranded cDNA Synthesis Kit (Thermo Fisher Scientific, K2561) following the manufacturer's protocol. Initially, 2–3 µg of RNA was combined with 1 µL of oligo(dT)18 (100 pmol) and 1 µL of 10 mM dNTPs, reaching a final volume of 11 µL with RNase-free water. After incubating at 65 °C for 5 minutes and snap-cooling on ice, a master mix consisting of 4 µl 5 x RT Buffer, 1 µL RNaseOUT, 3 µL Nuclease-free water, and 1 µL Maxima H Minus Reverse Transcriptase per sample was prepared. Incubation for 30 min at 50 °C followed, and the reaction was terminated by heating at 85 °C for 5 minutes. For the second strand synthesis, a master mix with 17.5 µL nuclease-free water, 10 µL of 5 X second strand reaction buffer, and 2.5 µL of second strand enzyme mix per sample was supplemented to the 20 µL first strand cDNA synthesis reaction. Samples were incubated at 16 °C for 60 min. Subsequently, 10 µL (100 U) of RNase I was added, and purification using AMPure XP beads-based (Agencourt, A63881) method was performed with a bead-to-sample ratio of 0.8 X, eluting in 21 µL of nuclease-free water. Concentrations of second-strand cDNA samples were determined using Qubit Fluorometric Quantitation (1 µL). Following this, end-prepping was conducted with NEBNext Ultra II End Repair / dA-tailing Module (NEB, cat # E7546). A mixture of 20 µL dscDNA sample, 22 µL nuclease-free water, 5.5 µL Ultra II End-prep reaction buffer, and 2.5 µL Ultra II End-prep enzyme mix was incubated at 20 °C for 15 min and 65 °C for 10 min. Cleanup with 1×AMPure XP Beads was performed, and elution was carried out in 10 µL nuclease-free water. Barcoding was achieved using Native Barcoding Expansion 1–12 (EXP-NBD104, ONT) by supplementing each sample with 2.5 µL Native Barcode and 10 µL Blunt/TA Ligase Master Mix, reaching a final volume of 22.5 µL. After incubation at RT for 20 min, 2 µL of EDTA was added to each sample to stop the reaction. Barcoded samples were pooled and purified using 0.7 X AMPure XP Beads and eluted in 31 µL nuclease-free water. The concentration of the pooled samples was determined. For adapter ligation, 5 µL NA, 10 µL NEBNext Quick Ligation Reaction Buffer (5 X), and 5 µL Quick T4 DNA Ligase (NEB, cat # E6056) were mixed and incubated in RT for 20 min. Lastly, the library was purified with 0.7 X AMPure XP Beads in a final elution volume of 33 µL EB. Concentration of the pooled barcoded library was determined using Qubit (1 µL). Finally, the library was mixed with sequencing buffer and loading beads before loading onto a primed R10.4.1 PromethION flow cell or R9.4.1 MinION flow cell.

## Nanopore-seq and NanopoReaTA data collection

Reads were basecalled using Guppy basecaller version 3.6.1 in high-accuracy (hac) mode for PromethION sequencing and super-accuracy for MinION sequencing. For a detailed overview of NanopoReaTA's requirements, pipelines, and additional tools, please refer to *Wierczeiko et al., 2023* or visit https://github.com/AnWiercze/NanopoReaTA. In this study, upon sequencing initiation, NanopoReaTA was activated following the guidelines outlined in the 'Step-by-Step Use of NanopoReaTA'. Two PromethION and one MinION flow cells were employed for this investigation. The cell culture samples, comprising a total of eight barcodes (barcodes 1–8), were loaded onto the first flow cell, and data were continuously collected over a 24 hr period. For the HEK293 and HeLa experimental setup, the samples were loaded onto a MinION flow cell, and data collection took place over a 72 hr sequencing period. The yeast samples were loaded onto a separate PromethION flow cell. Yeast setup 1 (barcodes 1–12) was initially loaded, and data collection extended for 24 hr. Following this period, sequencing was halted, and the PromethION flow cell was washed using the Flow Cell Wash Kit (EXP-WSH003, ONT). Subsequently, Yeast setup 2 (barcodes 1–3, 13–24) was loaded, and data were collected over another 24 hr period. For all experimental setups (cell culture and yeast), data points were collected at 1 hr, 2 hr, 5 hr, 10 hr, and 24 hr post-sequencing initiation (PSI). DEG overlap between the distinct time points, database, or sequencing devices was performed using Venn diagram web tool (https://bioinformatics.psb.ugent.be/webtools/Venn/). The collected data included general overview metrics, including the number of detected genes, gene variability, individual and combined read length distribution, as well as the usage timings of tools applied by

NanopoReaTA. Additionally, detailed information on differential gene and transcript expressions, including PCA, volcano plots, sample-to-sample distance plots, and heatmaps, was organized in the Figure Supplements.

## Acknowledgements

This work was funded by Deutsche Forschungsgemeinschaft (DFG, German Research Foundation) [project number 439669440 TRR319 RMaP TP B05 to MLW, TP A05/C01/C03 to MH; Project number 255344185 SPP1784, Startup Funding to MLW]. TB and SG acknowledge funding from the Emergent AI Center funded by the Carl-Zeiss-Stiftung. MLW and SG acknowledge funding from the Forschungsinitiative Rheinland-Pfalz and the ReALity initiative of the Johannes Gutenberg University Mainz. SG acknowledges funding by SFB 1551 Project No. 464588647 of the Deutsche Forschungsgemeinschaft (DFG).

## Additional information

### Funding

| Funder | Grant reference number | Author |
| --- | --- | --- |
| Deutsche Forschungsgemeinschaft | 439669440 | Stefan Mündnich Mark Helm Marie-Luise Winz |
| Deutsche Forschungsgemeinschaft | 255344185 | Mark Helm Marie-Luise Winz |
| Deutsche Forschungsgemeinschaft | 464588647 | Susanne Gerber |
| Deutsche Forschungsgemeinschaft | TRR319 RMaP TP B05 | Marie-Luise Winz |
| Deutsche Forschungsgemeinschaft | TP A05/C01/C03 | Mark Helm |
| Deutsche Forschungsgemeinschaft | TRR319 RMaP TP A07 | Susanne Gerber |
| Emergent AI Center | | Tamer Butto Susanne Gerber |
| Forschungsinitiative Rheinland-Pfalz | | Marie-Luise Winz Susanne Gerber |
| ReALity initiative of the Johannes Gutenberg University Mainz | | Marie-Luise Winz Susanne Gerber |
| Deutsche Forschungsgemeinschaft | TRR319 RMaP TP C04 | Susanne Gerber |

The funders had no role in study design, data collection and interpretation, or the decision to submit the work for publication.

### Author contributions

Tamer Butto, Conceptualization, Data curation, Formal analysis, Supervision, Investigation, Visualization, Methodology, Writing – original draft, Writing – review and editing; Stefan Pastore, Data curation, Software, Formal analysis, Investigation, Methodology, Writing – original draft, Writing – review and editing; Max Müller, Formal analysis, Investigation, Methodology; Kaushik Viswanathan Iyer, Marko Jörg, Julia Brechtel, Stefan Mündnich, Investigation, Methodology; Anna Wierczeiko, Software, Investigation, Writing – review and editing; Kristina Friedland, Mark Helm, Supervision, Investigation, Writing – review and editing; Marie-Luise Winz, Supervision, Investigation, Methodology, Writing – original draft, Writing – review and editing; Susanne Gerber, Conceptualization, Resources, Supervision, Investigation, Writing – original draft, Project administration, Writing – review and editing

Author ORCIDs
Tamer Butto ![ORCID] https://orcid.org/0000-0001-8028-0038
Marko Jörg ![ORCID] https://orcid.org/0009-0004-5799-1172
Mark Helm ![ORCID] https://orcid.org/0000-0002-0154-0928
Susanne Gerber ![ORCID] https://orcid.org/0000-0001-9513-0729

Reviewer #2 (Public review): https://doi.org/10.7554/eLife.98768.3.sa1
Author response https://doi.org/10.7554/eLife.98768.3.sa2

---

# Additional files

## Supplementary files
MDAR checklist

Supplementary file 1. Step-by-step guideline to NanopoReaTA.

## Data availability
All raw and processed sequencing data generated in this study have been submitted to the NCBI BioProject database (https://www.ncbi.nlm.nih.gov/bioproject/) under accession number PRJNA1090486.

The following dataset was generated:

| Author(s) | Year | Dataset title | Dataset URL | Database and Identifier |
|---|---|---|---|---|
| Butto T, Pastore S, Müller M, Iyer KV, Jörg M, Brechtel J, Mündnich S, Wierczeiko A, Friedland K, Helm M, Winz ML, Gerber S | 2024 | Real-time transcriptomic profiling in distinct experimental conditions | https://www.ncbi.nlm.nih.gov/bioproject/?term=PRJNA1090486 | NCBI BioProject, PRJNA1090486 |

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

## Appendix 1

## Optimizing double-stranded cDNA conversion and loading for Nanopore-Seq

During the experimental process, the conversion of double-stranded cDNA (dscDNA) and its subsequent loading into the flow cell were integral steps. Initially, the cDNA library preparation involved using the direct cDNA sequencing kit (SQK-DCS109), but it has since been discontinued. Therefore, we employed an alternative approach utilizing the Maxima H Minus Double-Stranded cDNA Synthesis Kit from Thermo Fisher Scientific (K2561), as outlined in the materials and methods. For the HEK and HeLa experiments, NanopoReaTA analysis was assessed on both PromethION (10-replicate and 2-replicate setups) and MinION (2-replicate setup) flow cells (R10 and R9, respectively). The PromethION experimental setup included a 24 hour sequencing period, with data collection intervals at 1 hr, 2 hr, 5 hr, 10 hr, and 24 hr. In the MinION experimental setup, sequencing extended for 72 hours.

During dscDNA library preparation, two strategies were implemented to evaluate sequencing efficiency for each condition. In the PromethION 10-replicate setup, 300 ng of dscDNA from HEK293 and HeLa cells were used (*Appendix 1—table 1*), while the 2-replicate setup utilized 400 ng of dscDNA (*Appendix 1—table 2*). This variation in cDNA input was based on the total amount of dscDNA required for optimal nanopore sequencing using the PromethION flow cell. For the MinION setup (two replicates), all available converted dscDNA for HEK293 (~850 ng) and HeLa cells was used (*Appendix 1—table 1*).

Upon utilizing NanopoReaTA's 'number of detected gene' utility, we noticed differences in the total number of detected genes between the different setups and flow cells. The 10-replicate PromethION setup showed a relatively balanced number of detected genes (*Appendix 1—figure 1A*). However, in the 2-replicate PromethION setup, the number of detected genes was lower in HEK293 compared to HeLa despite loading similar amounts of cDNA (*Appendix 1—figure 1B*). In contrast, the MinION setup, which utilized all converted cDNA per replicate, demonstrated a more consistent gene detection rate between conditions (*Appendix 1—figure 1C*).

Similarly, the cDNA conversion efficiency differed between RiboM and RiboP samples (*Appendix 1—table 3*). While 400 ng of cDNA was taken for the library preparation, RiboM samples did not reach this threshold; hence, all cDNA was utilized. Nonetheless, analysis using NanopoReaTA's 'number of detected gene' tool revealed that RiboM samples exhibited the highest number of detected genes compared to TotalR and RiboP (*Appendix 1—figure 1D*).

These observations raise the question of whether loading a similar amount of cDNA is a suitable strategy for generating even counts between distinct conditions and replicates, or if loading all the converted cDNA into the flow cell, leading to a more even number of reads generated by all replicates, would be preferable. While NanopoReaTA's analysis tools handle read normalization in both cases, addressing these differences is essential for refining best practices in long-read Nanopore RNA-seq experiments.

**Appendix 1—table 1.** HEK293 versus HeLa experimental setup (10 replicates per condition). Total amount of cDNA converted from 2 µg RNA and utilized in PromethION and MinION sequencing.

| | | HEK293 | | | | | | | | | | HeLa | | | | | | | | | | cDNA taken for sequencing |
|---|---|---|---|---|---|---|---|---|---|---|---|---|---|---|---|---|---|---|---|---|---|---|
| | | R1 | R2 | R3 | R4 | R5 | R6 | R7 | R8 | R9 | R10 | R1 | R2 | R3 | R4 | R5 | R6 | R7 | R8 | R9 | R10 | |
| *Total amount of cDNA generated (ng)* | PromethION | 516 | 496 | 480 | 516 | 508 | 512 | 464 | 480 | 456 | 532 | 348 | 404 | 424 | 420 | 508 | 492 | 440 | 468 | 488 | 432 | 300 ng (both HEK293 and HeLa) |

**Appendix 1—table 2.** HEK293 versus HeLa experimental setup (two replicates per condition). Total amount of cDNA converted from 2 µg RNA and utilized in PromethION and MinION sequencing.

| | | HEK293 | | HeLa | | cDNA taken for sequencing |
|---|---|---|---|---|---|---|
| | | R1 | R2 | R1 | R2 | |
| Total amount of cDNA generated (ng) | PromethION | 1,116 (bc1) | 1,172 (bc2) | 492 (bc3) | 440 (bc4) | 400 ng (both HEK293 and HeLa) |
| | MinION | 888 (bc4) | 824 (bc5) | 452 (bc6) | 564 (bc7) | All cDNA per replicate |

**Appendix 1—table 3.** rRNA-depleted and rRNA-enriched transcripts setup. Total amount of cDNA converted from 2 µg RNA (500 ng for RiboM) and utilized in PromethION sequencing.

| | Total R | | Ribominus | | Riboplus | |
|---|---|---|---|---|---|---|
| | R1 | R2 | R1 | R2 | R1 | R2 |
| Total amount of cDNA generated (ng) | 1,116 (bc1) | 1,172 (bc2) | 132 (bc5) | 163 (bc6) | 364 (bc7) | 2,120 (bc8) |
| cDNA taken for sequencing | 400 ng | 400 ng | 132 ng | 163 ng | 364 ng | 400 ng |

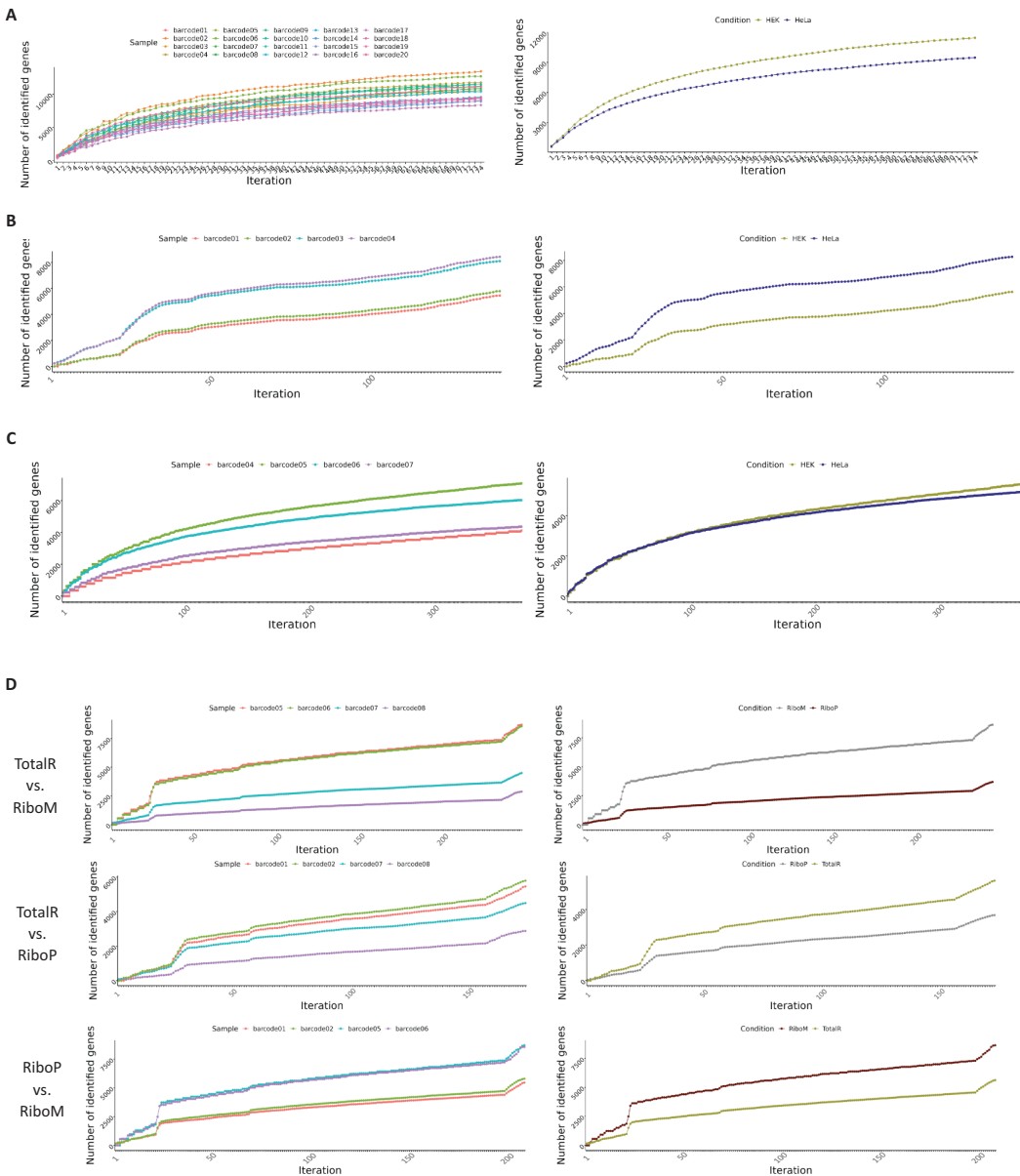

**Appendix 1—figure 1.** Comparison of # genes detected in different experimental conditions. (**A**) PromethION 10 replicate experimental setup. (**B**) PromethION 2 replicate experimental setup (**C**) MinION 2 replicate experimental setup (**D**) TotalR/Ribominus/Riboplus comparison. The plots present the number of identified genes after each iteration per sample (left) and per condition (right).

# Appendix 2

## Pooling of cDNA libraries after barcoding

In the yeast experimental setups, we observed low, yet noticeable abundance of deleted genes by examining the normalized read count. This concerned notably *NEW1* in Yeast setup 1 and *JLP2* in Yeast setup 2. Additionally, we detected the presence of antibiotic resistance genes in WT conditions where the expression of *KanR* or *HygR* should not have been present. Evidence of this discrepancy can be seen in the IGV snapshot, where a few reads were captured in conditions where they should not be expressed (*Appendix 2—figure 1*, see figure below). As one example, in Yeast setup 1, *HygR* was not detected in any sample, since none of the strains expressed HygR within this sequencing setup. However, in yeast setup 2, the identical WT-pEV(*HIS3*) (an aliquot of the identical RNA sample used in setup 1) was included in the library preparation and sequenced alongside the *rkr1Δ::HphMX* strains expressing *HygR*. As a result, we observed a few reads for *HygR* that mapped to the WT-pEV(*HIS3*) samples. This strongly suggests that detection of sequences that should not be present within strains is due to barcode cross-contamination during library preparation (Explanation described below). The same phenomenon can be seen for all other genes mentioned above. Only for the high detection of *HIS3* in all samples of setup 2 is there a different reason: The WT strain BY4741 bears the *his3Δ1* allele, which contains an internal deletion of 187 bp within the *HIS3* gene (*Daniel et al., 2006*). Beyond this internal deletion, which is clearly visible in the figure below, the remainder of the gene is still expressed at high levels and therefore detected and aligned to the *HIS3* gene, which is expected in this case.

We believe that one possible explanation of these discrepancies is potential barcode mix-up that occurs during library preparation in the sample barcoding step. During the barcoding stage, a Blunt/TA Ligase is used to ligate the barcodes to the dscDNA (for 20 min). The following step includes incubation of the sample with EDTA (supplied by the kit, EXP-NBD104, ONT) to inhibit the reaction and pooling all the barcodes together. We think that during this step of pooling, despite samples being supplemented with EDTA, it could be that some samples are still barcoded by other barcodes from the pool during mixing and therefore, we see a small, yet visible detection of genes that should not be present in the group. EDTA enzymatic inhibition occurs in a time-dependent manner (*Gonzalvo et al., 1997*), whereas in the protocol these steps occur relatively quickly with no longer incubation of samples with EDTA. Thus, we suggest two strategies to overcome this challenge. 1. The samples could be incubated with EDTA for at least 20 min to ensure proper inhibition of the enzymatic reaction so they will not cross to other samples. 2. Alternatively, each sample could be individually isolated after the barcoding step using Ampure XP bead purification. However, this approach carries the risk of material loss due to the small amount of cDNA being handled. Nonetheless, leveraging the rapid detection capabilities of NanopoReaTA, we quickly identified these discrepancies, showcasing its potential as an effective quality control tool for various experimental setups.

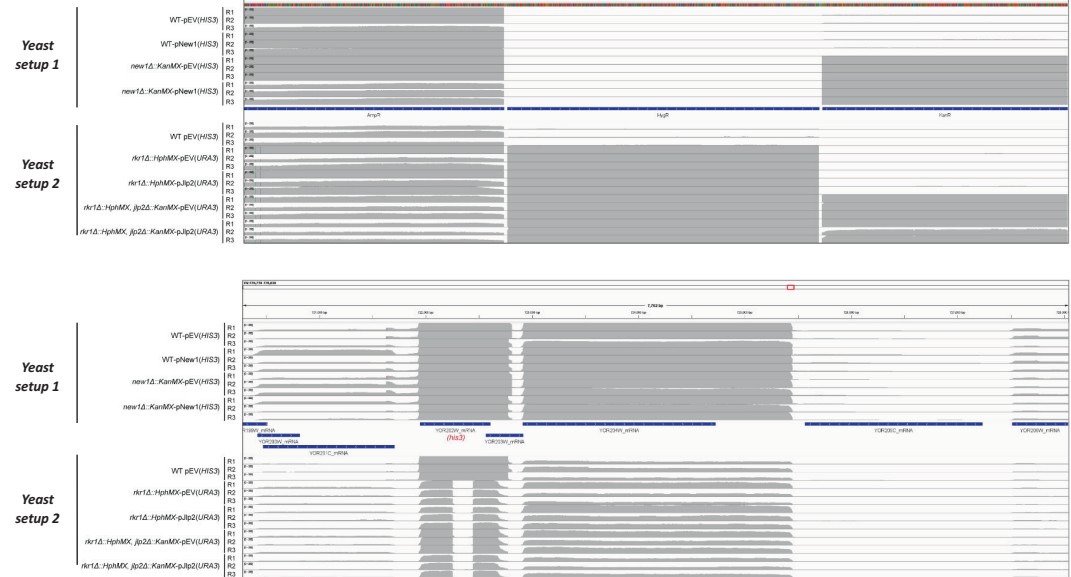

**Appendix 2—figure 1.** IGV coverage snapshots of selected gene loci in yeast setup 1 and 2. Top, IGV snapshots of reads mapping to AmpR, HygR, and KanR. Bottom, IGV snapshots of reads mapping to HIS3 (YOR202W).

