## [Editor Report · eLife Assessment]

This **useful** study presents a real-time transcriptomics analysis, with the aim of providing rapid access to sequenced data to reduce the costs associated with Oxford Nanopore long-read technology. The revised manuscript demonstrates the utilities with four sets of experiments with **convincing** evidence.

---

## [Referee Report · Reviewer #2 (Public review)]

Summary:

Transcriptomics technologies play crucial roles in biological research. Technologies based on second-generation sequencing, such as Illumina RNA-seq, encounter significant challenges due to the short reads, particularly in isoform analysis. In contrast, third-generation sequencing technologies overcome the limitation by providing long reads, but they are much more expensive. The authors present a useful real-time strategy to minimize the cost of RNA sequencing with Oxford Nanopore Technologies (ONT). The revised manuscript demonstrates the utilities with four sets of experiments with convincing evidence: (1) comparation between two cell lines; (2) comparison of RNA preparation procedures; (3) comparation between heat-shock and control conditions; (4) comparison of genetic modified yeast strains. The strategy will probably guide biologists to conduct transcriptomics studies with ONT in a fast and cost-effective way, benefiting both fundamental research and clinical applications.

Strengths:

The authors have recently developed a computational tool called NanopoReaTA to perform real-time analysis when cDNA/RNA samples are sequencing with ONT (Wierczeiko et al., 2023). The advantage of real-time analysis is that sequencing can be terminated once sufficient data has been collected to save cost. In this study, the authors demonstrate how to perform comprehensive quality control during sequencing. Their results indicate that the real-time strategy is effective across different species and RNA preparation methods. The revised manuscript addresses most of the major and minor limitations identified in the previous version, including: (1) explicitly detailing the methodology for isoform analysis and presenting the corresponding results; (2) increasing sample sizes and providing a clear explanation of related considerations; (3) clarifying the issue of sequential analysis; and (4) incorporating a new heat-shock experiment that better reflects real-world biological research.

Weaknesses:

A key advantage of RNA sequencing using ONT is its ability to facilitate isoform analysis. The primary strength of real-time analysis lies in its potential to reduce costs for researchers while enabling significant biological discoveries related to isoforms. Although the authors explicitly describe their approach to isoform analysis and introduce a new experiment in the revised manuscript, the study still lacks a concrete example that clearly demonstrates the substantial impact of their tool and strategy. While such an example may be beyond the intended scope of the current work, its absence limits a better assessment of the significance of the findings. Because the evaluation of a methodological approach ultimately depends on the additional scientific value it provides in research. It is possible that the full potential of this tool will be demonstrated in future studies by the authors or other researchers.

Furthermore, while the tool integrates a set of state-of-the-art methods, it does not introduce any novel methods. Consequently, the strength of evidence can be raised to "convincing".

---

## [Author Response]

The following is the authors’ response to the original reviews.

**Reviewer #1 (Public Review):**
In this study, the authors developed three case studies:(1) transcriptome profiling of two human cell cultures (HEK293 and HeLa)(2) identification of experimentally enriched transcripts in cell culture (RiboMinus and RiboPlus treatments)(3) identification of experimentally manipulated genes in yeast strains (gene knockouts or strains transformed with plasmids containing the deleted gene for overexpression). Sequencing was performed using the Oxford Nanopore Technologies (ONT), the only technology that allows for real-time analysis. The real-time transcriptomic analysis was performed using NanopoReaTA, a recent toolbox for comparative transcriptional analyses of Nanopore-seq data, developed by the group (Wierczeiko and Pastore et al. 2023). The authors aimed to show the use of the tool developed by them in data generated by ONT, evidencing the versatility of the tool and the possibility of cost reduction since the sequencing by ONT can be stopped at any time since enough data were collected.Strengths:Given that Oxford Nanopore Technologies offers real-time sequencing, it is extremely useful to develop tools that allow real-time data analysis in parallel with data generation. The authors demonstrated that this strategy is possible for both human cell lines and yeasts in the case studies presented. It is a useful strategy for the scientific community, and it has the potential to be integrated into clinical applications for rapid and cost-effective quality checks in specific experiments such as overexpression of genes.Weaknesses:In relation to the RNA-Seq analyses, for a proper statistical analysis, a greater number of replicates should have been performed. The experiments were conducted with a minimal number of replicates (2 replicates for case study 1 and 2 and 3 replicates for case study 3).

We have addressed this issue by performing two new sets of experiments: similar HEK293 vs HeLa with 10 replicates per condition and heatshocked vs non-heat shock with 6 replicates per condition. In the case of HEK293 vs HeLa comparison, we kept the 2 replicates per condition comparison to demonstrate the effect of limited replication number, simulating an early-stage evaluation of the experimental approach to obtain valuable quality control metrics. Nevertheless, we show that relevant and reproducible data can be obtained even with a lower replication number (2 replicates per condition), compared to a higher replication number (10 replicates), across both PromethION and MinION sequencing platforms.

Regarding the experimental part, some problems were observed in the conversion to doublestranded and loading for Nanopore-Seq, which were detailed in Supplementary Material 2. This fact is probably reflected in the results where a reduction in the overall sequencing throughput and detected gene number for HEK293 compared to HeLa were observed (data presented in Supplementary Figure 2). It is necessary to use similar quantities of RNA/cDNA since the sequencing occurs in real-time. The authors should have standardized the experimental conditions to proceed with the sequencing and perform the analyses.

We completely agree with the reviewer. In the 10-replicate HEK vs HeLa experiment, we collected similar data to what was presented in Supplementary Material 2. We chose to include this information to highlight the experimental variability that can arise during Nanopore-seq library preparation, particularly with cDNA synthesis. This type of information is not often highlighted in Nanoporebased studies, yet it is crucial to be aware of such differences. Despite these variations, we identified a consistent set of DEGs across comparisons of low versus high replicate numbers. Importantly, NanopoReaTA successfully provided realtime monitoring (e.g. detected number of genes per replicate/condition) as it allows for informed decision-making regarding the next steps in sequencing-based experiments.
**Reviewer #2 (Public Review):**
Transcriptomics technologies play important roles in biological studies. Technologies based on second-generation sequencing, such as mRNA-seq, face some serious obstacles, including isoform analysis, due to short read length. Third-generation sequencing technologies perfectly solve these problems by having long reads, but they are much more expensive. The authors presented a useful real-time strategy to minimize the cost of sequencing with Oxford Nanopore Technologies (ONT). The authors performed three sets of experiments to illustrate the utility of the real-time strategy. However, due to the problems in experimental design and analysis, their aims are not completely achieved. If the authors can significantly improve the experiments and analysis, the strategy they proposed will guide biologists to conduct transcriptomics studies with ONT in a fast and cost-effective way and help studies in both basic research and clinical applications.Strengths:The authors have recently developed a computational tool called NanopoReaTA to perform real-time analysis when cDNA/RNA samples are sequenced with ONT (Wierczeiko et al., 2023). The advantage of real-time analysis is that the sequencing can be stopped once enough data is collected to save cost. Here, they described three sets of experiments: a comparison between two human cell lines, a comparison among RNA preparation procedures, and a comparison between genetically modified yeasts. Their results show that the real-time strategy works for different species and different RNA preparation methods.Weaknesses:However, especially considering that the computational tool NanopoReaTA is their previous work, the authors should present more helpful guidelines to perform real-time ONT analysis and more advanced analysis methods. There are four major weaknesses:(1) For all three sets of experiments, the authors focused on sample clustering and gene-level differential expression analysis (DEA), and only did little analysis on isoform level and even nothing in any figures in the main text. Sample clustering and gene-level DEA can be easily and well done using mRNA-seq at a much cheaper cost. Even for initial data quality checking, mRNA-seq can be first done in Illumina MiSeq/NextSeq which is quick, before deep sequencing in HiSeq/NovaSeq. The real power of third-generation RNA sequencing is the isoform analysis due to the long read length. At least for now, PacBio Iso-seq is very expensive and one cannot analyze the data in real-time. Thus, the authors should focus on the real-time isoform analysis of ONT to show the advantages.

We are aware that isoform analysis is one of the powers of real-time monitoring of long-read data, especially with Nanopore-seq. That is why we have included pipelines such as DRIM-seq and DEX-seq, which could provide valuable information about the differential transcript usage (i.e. isoforms). However, interpreting the results in a biologically meaningful context, particularly regarding the role of specific isoforms, remains challenging. This is especially relevant as our main goal is to demonstrate NanopoReaTA's utility as a real-time transcriptomic tool that offers valuable quality control and meaningful insights. Nevertheless, in the heat-shock experiments, we have identified one isoform that was differentially expressed and included it in the main figure. We hope that with the right experimental setup, users could use the incorporated tools for meaningful analyses for isoforms identification.

(2) The sample sizes are too small in all three sets of experiments: only two for sets 1 and 2, and three for set 3. For DEA, three is the minimal number for proper statistics. But a sample size of three always leads to very poor power. Nowadays, a proper transcriptomics study usually has a larger sample size. Besides the power issue, biological samples always contain many outliers due to many reasons. It is crucial to show whether the real-time analysis also works for larger sample sizes, such as 10, i.e., 20 samples in total. Will the performance still hold when the sample number is increasing? What is the maximum sample number for an ONT run? If the samples need to be split into multiple runs, how the real-time analysis will be adjusted? These questions are quite useful for researchers who plan to use ONT.

We thank the reviewer for their suggestion. We performed the suggested experiment in the HEK293 vs HeLa, taking 10 replicates per condition and acquired the data during the sequencing. As you can see in the results (Figure 2), the performance held very well, from the first hour up until the 24hour mark. In theory, the maximum number of barcodes that can be integrated in a sequencing run can be used for the pair-wise comparison. We are using 24 barcoding kit (provided by ONT) therefore we can include up to 12 replicates per condition. We are aware that there is a 96 barcoding kit that could be used as well. However, it is important to note that with more samples integrated in the sequencing run, less reads will be generated per sample. Therefore, it is important to plan properly the number of replicates used per sequencing run.

(3) According to the manuscript, real-time analysis checks the sequencing data in a few time points, this is usually called sequential analysis or interim analysis in statistics which is usually performed in clinical trials to save cost. Care must be taken while performing these analyses, as repeated checks on the data can inflate the type I error rate. Thus, the authors should develop a sequential analysis procedure for real-time RNA sequencing.

We would like to respond to this comment by addressing two points: (1) Quality control: During the analysis we offer two main statistics, which enable scientists to assess the experimental development. For each iteration the change in relative gene counts per sample is computed to assess the convergence towards 0. Moreover, for each iteration the number of detected genes per sample is computed to assess whether the number of detected reads is saturated. These metrics allow the user to independently assess whether samples within the experimental development reach a stable state, to reveal a meaningful timepoint of data evaluation.

Sequential analysis: One solution to lower the type 1 error during sequential analysis is using the Pocock boundary, a systematic lowering of the p-value threshold depending on the number of interim analyses. We offer in NanopoReaTA a custom choice of the p-value threshold during the analysis. This allows researchers to set their parameters as needed.

(4) The experimental set 1 (comparison between two completely different human cell lines) and experimental set 2 (comparison among RNA preparation procedures) are not quite biologically meaningful. If it is possible, it is better for the authors to perform an experiment more similar to a real situation for biological discovery. Then the manuscript can attract more researchers to follow its guidelines.

We took the suggestion of reviewer 2 (from recommendation for authors) to perform heat-shock experimental comparison between heatshocked and non-heat shocked cells from the same cell line (HEK293). We sequenced the sample (6 replicates per condition) and one-hour postsequencing initiation, we already identified three DEGs (including HSPA1A, DNAJB1, and HSP90AA1) known to be upregulated in heat shock conditions (Yonezawa and Bono 2023, Sanchez-Briñas et al. 2023). Therefore, we illustrate how NanopoReaTA can capture biologically relevant insights in real time.

**Reviewer #1 (Recommendations for The Authors):**
(1) The comparison between two different human cell lines doesn't have much biological relevance. It would be more interesting and useful to evaluate the genes and transcripts expressed from the same cell in different conditions.

As mentioned previously, we conducted a heat-shock experimental comparison between heat-shocked and non-heat-shocked within the same cell line HEK293. We observed reliable results already within one hour of initiating the sequencing.

(2) Increase the number of replicates to give greater confidence in the results.

We have addressed the replicate issue by performing two new sets of experiments: HEK293 vs HeLa with 10 replicates per condition and heatshocked vs non-heat shock with 6 replicates per condition. In both cases, we obtained reliable and reproducible results (even when comparing with lower replicate number).

(3) One of the advantages of performing Nanopore sequencing is the possibility of sequencing RNA molecules directly. It would be interesting to test the real-time analysis strategy in parallel using direct RNA sequencing if it is possible.

That is a great point. In theory, it would be possible to perform realtime differential gene expression on direct RNA data (since the pipeline for such analysis is already integrated in NanopoReaTA), however the limiting factor is the lack of multiplexing. To perform real-time transcriptomic analysis with direct RNA-seq data, one would need to sequence at least 4 flow cells (MinION or PromethION), each containing one sample (2 flow cells per condition to perform pairwise transcriptomic analyses). Despite the possibility of such an analysis, this scenario will not be cost-effective as this will increase significantly the costs for the amount of data gathered. We are aware that ONT is planning to release a multiplexing option to direct RNA-seq in the unforeseen future. We have integrated the option of direct RNA-seq analyses for the day that such option will be available, and the users will be able to perform real-time transcriptomic analysis with dRNA-seq data.

Some minor weakneses are below:(4) With respect to the text as a whole, the authors should be more careful with standardization, such as mL/ml and uL/ul, Ribominus/RiboMinus.

We have standardized the nomenclature to µL, mL and Ribominus (due to trademark).

(5) Set up paragraphs on page 9 and throughout the text when necessary.

We have set the suggested paragraphs on page 9 and throughout the text.

(6) Please, check the word form in the sentence: "To isolate the RNA form theRiboMinus{trade mark, serif} supernatant.."

The word has been corrected.

(7) In order to make clear to the reader at the outset, I suggest including in the methodology how many biological replicates were performed for each cell type studied (cell lines and yeast strains).

_For cell line w_e have included now the number of replicates used for each replicate. We have included this also for yeast setups.

(8) Please, check the Supplementary Tables as the word VERDADEIRO has not been translated (TRUE) in Supplementary Table 1.

This issue appears to be influenced by the language settings configured on the viewer's computer.

(9) On page 17, I suggest including the absorbance used to measure RNA concentration in HEK293 and HeLa cell lines. Also, I suggest including how the quality of the RNA extracted from the cell cultures and yeast strains was determined. Was the ratio 260/280 and 260/230 calculated? Given that the material was extracted with Trizol, which has phenol and chloroform in its composition, it would be important to evaluate the quality of the RNA, especially by calculating the 260/230 ratio.

We have included a statement regarding the concentrations and quality of RNA in the “RNA isolation” section within the material and methods.

(10) On page 18, the topic of Selective purification of ribosomal-depleted (RiboMinus) and ribosomal-enriched (RiboPlus) transcripts needs to be better detailed, especially in the last two sentences. For example: "The pooled bead samples (containing the rRNA) were further processed with Trizol RNA isolation to complete the purification." This sentence should be detailed to make it clear that this procedure is what you call ribosomal-enriched (RiboPlus).Qualitative analysis of the material was performed after rRNA depletion and enrichment.

We have made these sentences clearer.

(9) On the topic of Direct cDNA-native barcoding Nanopore library preparation and sequencing, in the following sentences: "Concentration determination (1 μl) and adapter ligation using 5 μL NA, 10 μL NEBNext Quick Ligation Reaction Buffer (5X), and 5 μL Quick T4 DNA Ligase (NEB, cat # E6056) were performed. Pooled library purification with 0.7X AMPure XP Beads resulted in a final elution volume of 33 μl EB. Concentration of the pooled barcoded library was determined using Qubit (1 μl)."

Two concentration determinations were performed, before and after adapter ligation. I suggest writing one sentence for concentration determination and another for adapter ligation.

We applied the reviewer’s suggestion.

(11) In the section Experimental Design in Results, the first sentences are part of the methodology and are described in materials and methods. I suggest removing it from the results and rewriting the text. Results of the RNA extraction methodology and library preparation were shown in supplementary material. Thus, the authors could mention that the results were presented in supplementary material.

We have revised this section to remove the details of RNA extraction and library preparation, focusing instead on the pipeline and experimental setups. The methodology is outlined in Figure 1, as well as in the materials and methods and the supplementary figures for each experimental setup.

**Reviewer #2 (Recommendations For The Authors):**
For major weakness 4 described in the Public Review, the authors could try experiments like:(1) comparison between females and males of tissues or primary cells; or(2) comparison between cell lines before and after heat shock.They are easy to perform and much more similar to real experimental designs for discovery, and the authors may actually have some new findings because usually people do not do much investigation on the isoform level using mRNA-seq.

We thank the reviewer for their suggestions. We performed the heat-shock experimental comparison between heat-shocked and non-heat shocked cells from the same cell line (HEK293). We sequenced the sample (6 replicates per condition) and already one-hour post-sequencing initiation, we identified three DEGs including HSPA1A, DNAJB1, and HSP90AA1 reported to be upregulated heat shock conditions (Yonezawa and Bono 2023, Sanchez-Briñas et al. 2023). We have identified one differentially expressed isoform and included it in the main figure.

There are two minor weaknesses:(1) Many figure numbers in the main text are wrong, including:Page 4, "similarity plot and principal component analysis (PCA) (Figure 1B, 1C)";Page 7, "same intervals as mentioned earlier (Figure 1A)", and "Next, we inspected the PCA and dissimilarity plots (Figure 2B)";Page 10, "process (Supplementary Figure 19A) until the 24-hour PSI mark point (Figure 9B)", and "NEW1 was the sole differentially expressed gene (Figure 9D)".The authors should be more careful about this. It is very confusing for readers.

We have addressed these points in the text.

(2) The texts in the figures are too small to recognize, especially in Figures 4 and 5. The reason is that there are too many sub-figures in one figure. Is that really necessary to put more than 20 sub-figures in one? The authors should better summarize their results. For example, remove sub-figures with little information; do not show figures with the same styles again and again in the main text and just summarize them instead.

We thank the reviewer for the suggestion. We have updated the figure to focus on the most relevant comparisons (new1Δ-pEV vs. WT-pEV and rkr1Δ-pEV vs. WT-pEV), providing a clearer and more realistic comparison between mutant and wild-type conditions in the main figure. Additionally, a summary and all related comparisons are included in Supplementary Documents S4 and S5. We believe these supplementary figures are essential to demonstrate NanopoReaTA's capabilities as a quality control tool, effectively detecting expected transcriptomic alterations in real-time.